# CD39 expression by regulatory T cells participates in CD8+ T cell suppression during experimental *Trypanosoma cruzi* infection

**Cintia L. Araujo Furlan**[1,2]*, **Santiago Boccardo**[1,2], **Constanza Rodriguez**[1,2], **Verónica S. Mary**[1,2], **Camila M. S. Gimenez**[1,2], **Simon C. Robson**[3], **Adriana Gruppi**[1,2], **Carolina L. Montes**[1,2], **Eva V. Acosta Rodríguez** [1,2]*

**1** Departamento de Bioquímica Clínica, Facultad de Ciencias Químicas, Universidad Nacional de Córdoba, Córdoba, Argentina, **2** Centro de Investigaciones en Bioquímica Clínica e Inmunología, CONICET, Córdoba, Argentina, **3** Center for Inflammation Research, Department of Anesthesia, Beth Israel Deaconess Medical Center, Harvard Medical School, Boston, Massachusetts, United States of America

* claraujo@unc.edu.ar (CLAF); eva.acosta@unc.edu.ar (EVAR)

**Data Availability Statement:** All data provided within the manuscript and supporting information.

## Abstract

An imbalance between suppressor and effector immune responses may preclude cure in chronic parasitic diseases. In the case of *Trypanosoma cruzi* infection, specialized regulatory Foxp3+ T (Treg) cells suppress protective type-1 effector responses. Herein, we investigated the kinetics and underlying mechanisms behind the regulation of protective parasite-specific CD8+ T cell immunity during acute *T. cruzi* infection. Using the DEREG mouse model, we found that Treg cells play a role during the initial stages after *T. cruzi* infection, restraining the magnitude of CD8+ T cell responses and parasite control. Early Treg cell depletion increased the frequencies of polyfunctional short-lived, effector T cell subsets, without affecting memory precursor cell formation or the expression of activation, exhaustion and functional markers. In addition, Treg cell depletion during early infection minimally affected the antigen-presenting cell response but it boosted CD4+ T cell responses before the development of anti-parasite effector CD8+ T cell immunity. Crucially, the absence of CD39 expression on Treg cells significantly bolstered effector parasite-specific CD8+ T cell responses, preventing increased parasite replication in *T. cruzi* infected mice adoptively transferred with Treg cells. Our work underscores the crucial role of Treg cells in regulating protective anti-parasite immunity and provides evidence that CD39 expression by Treg cells represents a key immunomodulatory mechanism in this infection model.

## Author summary

Chagas disease, caused by *Trypanosoma cruzi*, can result in severe health complications. While the exact mechanisms underlying the disease's pathogenesis remain incompletely understood, the host's inflammatory immune response is believed to play a critical role. To shed light on disease mechanisms and potential treatments, we investigated the impact of regulatory T (Treg) cells on the development of effector immune responses against *T.*

**Funding:** This work was supported by: Agencia Nacional de la Investigación, el Desarrollo Tecnológico y la Innovación (https://www.argentina.gob.ar/ciencia/agencia) under grants PICT 2018-01791 and PICT 2020-0487 awarded to EVAR, and the National Institute of Allergy and Infectious Diseases of the National Institutes of Health (https://www.niaid.nih.gov/) under Award Number R01AI169482 to EVAR. The content is solely the responsibility of the authors. Funders play no role in the study design, data collection and analysis, decision to publish, or preparation of the manuscript.

**Competing interests:** The authors have declared that no competing interests exist.

*cruzi*. Our findings reveal that Treg cells dampen parasite-specific CD8+ T cells, a crucial arm of the immune response in counteracting the parasite. Notably, this regulatory influence occurs primarily during the early stages of *T. cruzi* infection. Furthermore, we observed that while Treg cells have minimal effects on antigen-presenting cells, they modulate the magnitude and phenotype of conventional CD4+ T cells. Importantly, we identified CD39, a molecule involved in the purinergic pathway, as essential for the suppressive functions of Treg cells during *T. cruzi* infection. Our findings enhance the understanding of the regulatory response during the acute phase of *T. cruzi* infection and may have implications for the development of novel therapeutic strategies.

## Introduction

The protozoan parasite *Trypanosoma cruzi* is the causative agent of Chagas disease, a neglected tropical disease, which is endemic to South and Central America, with some regions of North America also impacted. Due to human migration, Chagas disease has become a worldwide concern [1]. Early detection of infection is challenging and access to curative, non-toxic pharmacological treatment is limited, often resulting in lifelong disease for most individuals. During the acute phase, *T. cruzi* undergoes intensive replication, becoming detectable in the bloodstream and then spreading systemically to host tissues. The infection triggers a strong type 1 pro-inflammatory response, involving activation of cells from both the innate and adaptive immune system [2].

Macrophages, neutrophils, natural killer cells and dendritic cells act as the first defense against *T. cruzi*, while specific T and B cell responses develop over time. Notably, highly immunodominant parasite-specific CD8+ T cell responses play a crucial albeit delayed role in the optimal control of parasite replication [3,4]. Immunological help from CD4+ T cells is required to achieve robust parasite-specific CD8+ T cell responses [5]. However, although effector mechanisms are essential for limiting systemic acute infection, the immune response often fails to completely eliminate *T. cruzi*, allowing pathogen persistence in target tissues and leading to chronic disease. While these findings are primarily derived from experimental models, human studies have corroborated the importance of robust Th1 and CD8+ T cell responses in controlling *T. cruzi* infection during the acute phase [2,6]. Once the parasite is controlled and the chronic phase arises, a substantial proportion of patients remain asymptomatic but approximately 30–40% of infected individuals develop life-threatening cardiac and/or gastrointestinal complications.

Regulatory T (Treg) cells are a distinct subset of CD4+ T cells that are characterized by the expression of the transcription factor Foxp3, which equips them with the remarkable capacity to exert suppressive effects on a wide range of immune cell populations [7]. During infectious inflammatory responses, Treg cell-mediated immunity may develop once the infection has been controlled to restore homeostasis and/or prevent collateral tissue damage. Conversely, pathogens can also induce Treg cells, as an early mechanism of immune evasion. As a result, Treg cells can have both beneficial and detrimental effects during infections, demanding a delicate and kinetic balance between effector and regulatory responses to achieve pathogen clearance, while protecting against excessive inflammation [8]. Treg cell accumulation has been observed in peripheral blood, secondary lymphoid organs and target tissues of chronic bacterial, viral and parasitic infections in both mice and humans [7,9]. On the other hand, decreased numbers of Treg cells have been reported during the acute phase of a few infections, such as those caused by *Toxoplasma gondii* [10,11], *Listeria monocitogenes* [10], vaccinia virus [10],

LMCV clone Armstrong [12], and more recently, Chikungunya virus [13]. These findings underscore the complexity of Treg cell dynamics and illustrate the context-dependent roles in different infectious scenarios.

Using an experimental model, we have previously described a relative decrease in Treg cell numbers during acute *T. cruzi* infection, which did not fully recover in the transition to the chronic phase [14]. Additionally, we found that activated Treg cells exhibit phenotypic and transcriptional profiles consistent with the suppression of type 1 inflammatory responses. Our research also highlighted a crucial role of the natural contraction of Treg cell response during the acute *T. cruzi* infection in facilitating the development of protective anti-parasite CD8+ T cell immunity. In line with these observations, a study by Ersching *et al.* reported that Treg cells suppress CD8+ T cell priming in an immunization model using *T. cruzi*-stimulated dendritic cells [15]. Through the use of immune deficient mice and blocking antibody treatments, these authors have demonstrated that this effect was independent of IL-10 but partially mediated by CTLA-4 and TGF-β. However, the specific mechanisms employed by Treg cells to suppress effector responses during acute *T. cruzi* infection have yet to be fully elucidated. Importantly, current data on Treg cell responses in human Chagas disease are limited to the chronic phase, wherein an increased Treg cell frequency and functionality have been associated with less severe cardiomyopathy [16–19]. While these findings suggest a beneficial role for this cell subset during human chronic Chagas disease, where immunoregulation may be instrumental in reducing immunopathology, the situation may differ significantly in the acute phase, where excessive regulation could hinder the function of effector immune subsets.

CD39, also known as ectonucleoside triphosphate diphosphohydrolase 1 (ENTPDase1/gene *ENTPD1*), is a nucleotide-metabolizing enzyme involved in immune regulation and inflammation [20]. In response to pro-inflammatory stimuli, intracellular ATP is released into the extracellular space, acting as a danger signal that promotes inflammation. However, the presence of CD39-expressing "regulatory" cells modulate this process by catalyzing ATP to AMP conversion. In turn, AMP can be further converted into the immunoregulatory molecule adenosine (ADO) by CD73. Thus, the tandem activity of CD39 and CD73 holds the potential to shift the inflammatory environment towards an immunosuppressive state through the degradation of ATP into ADO [21]. CD39 expression has been identified in various immune cell populations, including Treg cells, in both mice and humans–as well as marking immune exhaustion [22–24]. Consequently, the investigation of Treg cells expressing CD39 in different inflammatory contexts has shed light on their significant contributions to autoimmune diseases, cancer, allergies, and viral infections [25,26]. However, the role of CD39+ Treg cells remains largely unexplored in the context of microbial and parasitic infections, warranting further investigation to elucidate their potential implications in such settings.

Herein, we aimed to comprehensively investigate the biological significance of Treg cells in the context of *T. cruzi* infection. We sought to determine the specific time window during which Treg cells exert immune suppression and to identify the immune cell populations that are particularly sensitive to these regulatory effects in the acute phase of this parasitic infection. Through our investigations, we uncovered that during the initial stages of infection, endogenous Treg cells play a role in suppressing the expansion of CD4+ T cells and the development of anti-parasite effector CD8+ T cell responses. Moreover, we discovered that CD39 serves as a relevant regulatory mechanism through which Treg cells mediate suppression of *T. cruzi*-specific CD8+ T cells. These findings offer valuable insights into the role of Treg cells during the acute phase of *T. cruzi* infection and warrant further investigation into their long-term effects during the chronic phase. Such exploration would be instrumental in assessing potential immunomodulatory strategies targeting Treg cells or CD39. While parasiticidal drugs remain

standard treatment, complementary immunomodulatory approaches could improve Chagas disease management.

## Results

### Treg cell depletion expands parasite-specific CD8+ T cells and improves parasite control during acute *T. cruzi* infection

We previously demonstrated that an exogenous increase in Treg cell numbers during acute *T. cruzi* infection dampens the anti-parasite effector response and subsequently influences infection control [14]. To further investigate the biological role of the endogenous Treg cell response and elucidate the underlying mechanisms involved in effector T cell suppression during this infection, we employed the DEREG (DEpletion of REGulatory T cells) mouse model [27] to conduct Treg cell depletion experiments.

As illustrated in Fig 1A, DEREG mice were infected and received diphtheria toxin (DT) injections at days 5 and 6 post-infection (pi), in order to target Treg cells before the emergence of adaptive anti-parasite responses that occurs at day 10 pi in our infection model [28]. We observed that the depletion strategy efficiently reduced Treg cell frequencies in the blood of DT-treated mice compared to PBS-treated controls, starting from at least day 11 pi (Fig 1B). A kinetics analysis in blood revealed that Treg cell frequencies returned to normal levels around 15 days after DT treatment in non-infected mice (Fig 1C). However, in DT-treated infected animals, Treg cell frequencies remained significantly decreased until day 33 pi compared to Treg-sufficient infected controls. Thus, DT injection effectively intensified and prolonged the natural contraction of the Treg cell response that we and others have previously reported for *T. cruzi* infection [3,14,29].

We further examined the spleen and the liver, a target tissue for *T. cruzi* that exhibits a substantial leukocyte infiltration during the acute phase of the infection [30–32]. Importantly, we observed a reduction in Treg cell frequencies following DT treatment in both the spleen and liver of infected DEREG mice throughout the acute phase, including day 20 pi, a time point when the most substantial natural contraction of the Treg cell response is detected in comparison to non-infected mice (Fig 1D and 1E). Similarly, a decrease in Treg cell frequency after treatment was noted within the CD4+ T cell gate (S1A and S1B Fig). Indeed, DT injection led to a significant decrease in Treg cell count in these organs at various dpi (Fig 1F).

Consistent with our previous report [14], we found that Treg-depleted infected animals showed reduced levels of blood parasitemia (Fig 2A) and tissue parasite burden in *T. cruzi* target tissues, such as heart and liver, compared to their control counterparts at day 20 pi (Fig 2B), which coincides with the peak of the CD8+ T cell response in our model [28,33]. However, no differences in parasite load were observed in the spleen (Fig 2B). Considering the critical role of CD8+ T cells in controlling *T. cruzi* replication [34], we quantified CD8+ T cells specific for the immunodominant epitope TSKB20 (*T. cruzi* trans-sialidase amino acids 569–576 –ANYKFTLV–) [35] in blood, spleen and liver (Fig 2C). Through kinetics studies in blood, we determined that Treg cell depletion significantly increased the frequencies of circulating TSKB20-specific CD8+ T cells from day 20 to day 33 pi in DT-treated animals compared to controls (Fig 2C and 2D). Subsequently, the response progressively contracted, reaching frequencies similar to PBS-treated mice at day 55 pi. Similar results were observed in the spleen and liver, where higher relative and absolute numbers of parasite-specific CD8+ T cells were observed in Treg-depleted animals compared to Treg-sufficient mice at day 20 pi (Fig 2E and 2F). We also observed increases in the frequency of total CD8+ T cells in the blood of DT-treated infected mice compared to controls at certain time points (S1C Fig). Additionally, total CD8+ T cell frequency was elevated in the liver at 20 dpi, while remaining unchanged in the

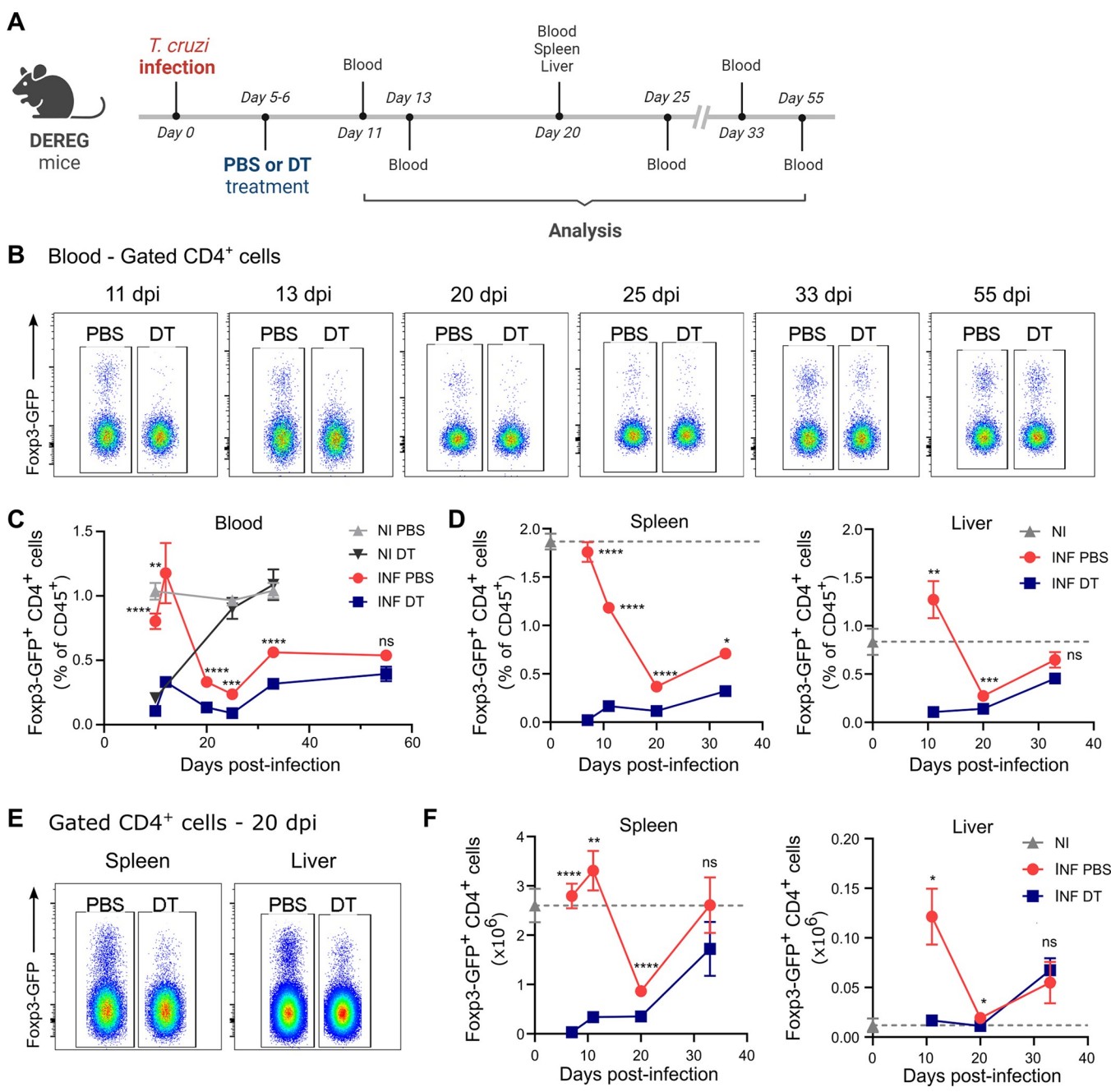

**Fig 1. DT injection efficiently depletes Treg cells in DEREG mice during acute *T. cruzi* infection. A)** Experimental scheme illustrating DT treatment (created with BioRender.com). **B-C)** Representative flow cytometry dot plots depicting Foxp3-GFP expression (B) and Treg cell frequencies (C) in the blood of PBS or DT-treated *T. cruzi*-infected (INF) DEREG mice at different days post-infection (dpi) and non-infected (NI) controls. **D-F)** Kinetics analysis of Treg cell frequencies (D), representative flow cytometry dot plots showing Foxp3-GFP expression (E), and kinetics analysis of Treg cell absolute numbers (F) in spleen and liver of PBS or DT-treated *T. cruzi*-infected DEREG mice and non-infected controls. All data are presented as mean ± SEM. Data were collected from 1–3 independent experiments. A total of 2–18 mice per group were included. In (C) n = 2–3 for NI groups, n = 12–13 at 10 dpi, n = 7 at 12 dpi, n = 16–18 at 20 dpi, n = 6 at 25 dpi, n = 12 at 33 dpi, n = 5–7 at 55 dpi. In (D) and (F) n = 2–7 for NI, n = 10–12 at 7 dpi, n = 3–4 at 11 dpi, n = 8–14 at 20 dpi, n = 4–5 at 33 dpi. Statistical significance was determined by Unpaired t test or Mann Whitney test, according to data distribution. Statistical analysis represents pairwise comparisons between INF PBS and INF DT groups. * P ≤ 0.05, ** P ≤ 0.01, *** P ≤ 0.001, **** P ≤ 0.0001 and ns = not significant.

spleen (S1D Fig). However, the absolute number of CD8+ T cells showed no differences after Treg cell depletion in both organs (S1E Fig).

It is important to note that the observed effects on parasite levels were not due to potential toxic effects of DT itself. This was highlighted by the fact that DT injection did not impact parasitemia or the frequency of parasite-specific CD8+ T cells in WT littermates (S2A and S2B Fig). Additionally, we verified that DT did not directly impact *T. cruzi in vitro*, as parasite survival remained unaffected after 24 hours of culture with increasing DT doses, including the theoretically achievable concentration of 5.2 nM during DT treatment *in vivo*. In contrast, benznidazole notably reduced parasite survival, as anticipated (S2C Fig).

Finally, we evaluated the impact of Treg cell depletion on tissue damage, observing no significant effect on the plasmatic levels of biochemical markers (S2D Fig), except for a tendency towards a reduction in GOT activity. This suggests that a more robust effector response is not necessarily linked to increased acute tissue damage.

Next, we studied whether the role of Treg cells in regulating the effector response was time-dependent during the acute phase of *T. cruzi* infection. To address this, we modified the DT injection schedule to days 11 and 12 pi (S3A Fig), aiming to deplete Treg cells at the time point when, as we have previously reported, the natural reduction in Treg cell frequency begins, and the anti-parasite T cell response is already detectable [14,28]. Surprisingly, despite Treg cells remained depleted in blood and spleen until at least day 21 pi (S3B and S3C Fig), the delayed Treg cell depletion strategy had no impact on parasitemia levels or the frequencies of *T. cruzi*-specific CD8+ T cells in the blood (S3D and S3E Fig). Furthermore, no effects were observed in the frequencies of the anti-parasitic response at day 21 pi in the spleen of DT-treated animals, as compared to the control group (S3F Fig).

Altogether, these results indicate that Treg cells play a role during the initial stages of *T. cruzi* infection, likely influencing the priming and/or activation of the effector CD8+ T cell response. The suppressive function of Treg cells during the early phase of infection ultimately impacts the magnitude of the TSKB20-specific CD8+ T cell response and, consequently, the control of parasite replication.

## Early depletion of Treg cells promotes differentiation of parasite-specific CD8+ T cells into polyfunctional short lived effector cells

To examine the influence of Treg cell depletion on specific subsets of CD8+ T cells in the context of *T. cruzi* infection, we investigated the contribution of effector, memory and activated cells. Using flow cytometry, we identified short-lived effector cells (SLEC) and memory precursor effector cells (MPEC) based on CD44, KLRG-1 and CD127 expression, as illustrated in S4A Fig [36,37]. Our analysis revealed that early Treg cell depletion increased the frequency of the SLEC subset within *T. cruzi*-specific CD8+ T cells from the spleen as well as the frequency and absolute numbers of SLEC within liver parasite-specific CD8+ T cells (Fig 3A and 3B). In contrast, the MPEC subset did not show significant changes in frequency or absolute numbers in parasite-specific CD8+ T cells from the spleen and liver after DT injection (Fig 3C). Within total CD8+ T cells, Treg cell depletion also resulted in increased frequency of SLEC cells in the spleen but not the liver, while MPEC remained the same in both organs (S4B Fig).

We further investigated the phenotypic characteristics of CD8+ T cell responses following Treg cell depletion in the spleen and liver. Our analysis revealed no significant differences in the frequencies of total or parasite-specific CD8+ T cells expressing various activation markers (CD69, CD25, CD44, and CD38), the regulatory molecule CD39, inhibitory receptors (PD-1, Lag-3, and Tim-3), or effector molecules (Granzyme A and B) between PBS- and DT-treated mice (S4C Fig).

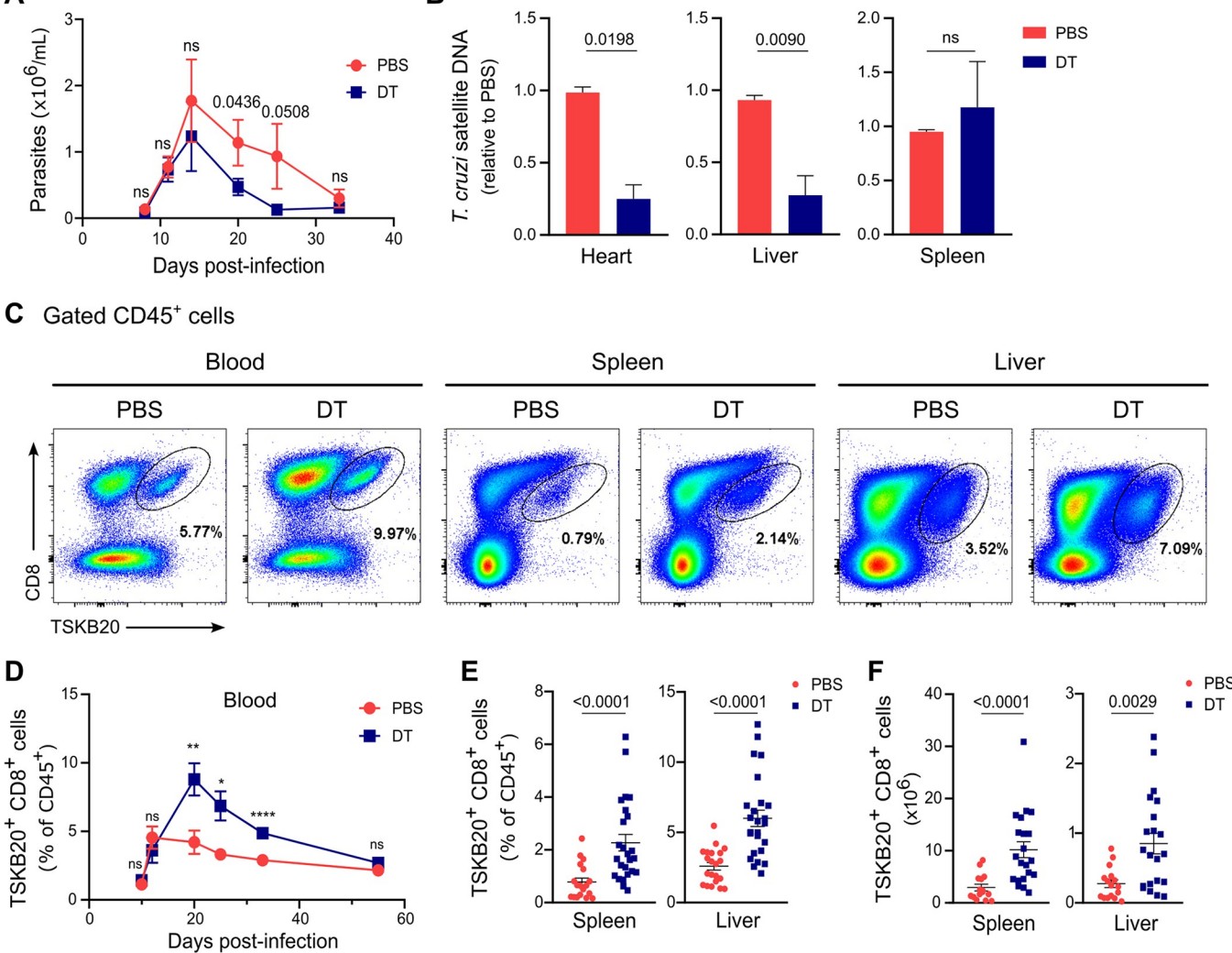

**Fig 2. Treg cell depletion increases *T. cruzi*-specific CD8+ T cell expansion and improves parasite control during acute infection. A)** Kinetics of blood parasite counts in PBS or DT-treated DEREG mice. **B)** Parasite load in heart, liver, and spleen of PBS or DT-treated DEREG mice at day 20 pi. Values were calculated using the ΔΔCT algorithm, with GAPDH utilized as a housekeeping control for normalization, and the sample of PBS-treated mice serving as a reference. **C-E)** Representative flow cytometry dot plots showing TSKB20-specific CD8+ T cell detection at day 20 pi (C), their frequency quantification in blood at different dpi (D), and in spleen and liver at day 20 pi (E) of PBS or DT-treated DEREG mice. **F)** Absolute numbers of TSKB20-specific CD8+ T cells in spleen and liver according to (E). All data are presented as mean ± SEM. In (A), (D), (E) and (F) data were collected from 1–3 independent experiments at most dpi, and 5–8 independent experiments at day 20 pi according to the analyzed tissue. In (B) data were pooled from 2–4 independent experiments. In (E-F) each symbol represents one individual mouse. A total of 4–42 mice per group were included. In (A) n = 11–12 at 8 dpi, n = 12 at 11 dpi, n = 11 at 14 dpi, n = 35–42 at 20 dpi, n = 7–9 at 25 dpi, n = 4–5 at 33 dpi. In (D) n = 12–13 at 10 dpi, n = 7 at 12 dpi, n = 16–18 at 20 dpi, n = 6 at 25 dpi, n = 12 at 33 dpi, n = 5–7 at 55 dpi. Statistical significance was determined by Unpaired t test or Mann Whitney test, according to data distribution. P values for pairwise comparisons are indicated in the graphs. * P ≤ 0.05, ** P ≤ 0.01, *** P ≤ 0.001, **** P ≤ 0.0001 and ns = not significant.

Based on our preceding results, we further investigated the functionality of parasite specific CD8+ T cells after DT treatment. To test this, we examined the degranulation ability by CD107a surface mobilization together with secretion of the effector cytokines IFN-γ and TNF upon *in vitro* specific and polyclonal stimulation (S5A Fig). As shown in Fig 3D, CD8+ T cells from 21 day-infected Treg-depleted mice exhibited an increased proportion of polyfunctional cells that produce two and three effector mediators (degranulation plus IFN-γ production with or without TNF release) when stimulated with TSKB20 but not after polyclonal

stimulation with PMA/Ionomycin. No significant changes were detected in other parasite-specific cells expressing one or two effector mediators (S5B Fig).

Altogether, our data indicate that early Treg cell depletion during *T. cruzi* infection primarily increases the magnitude of the parasite-specific CD8+ T cell response, likely favoring the differentiation of effector cells with polyfunctional features, without significant changes in other phenotypic or functional attributes.

## Effects of early Treg cell depletion on antigen-presenting cell populations and conventional CD4+ T cells

Next, we investigated whether other cell populations could be modulated by Treg cells at early stages of the infection and indirectly impact CD8+ T cell responses. Considering that only early depletion of Treg cells resulted in the increase of parasite-specific effector CD8+ T cells and improved control of parasite replication at day 20 pi, and taking into account that the TSKB20-specific CD8+ T cell response starts to arise around day 10 pi in our infection model (Fig 2D) [28], we hypothesized that Treg cells may play a role at early events of T cell priming during *T. cruzi* infection. To address this question, we evaluated the response of antigen-presenting cells (APCs) and innate immune cells the following day after DT treatment in the spleen of DT or PBS-treated and infected mice as well as non-infected controls, using multiparametric flow cytometry. As depicted in the Uniform Manifold Approximation and Projection (UMAP) visualization plots of Fig 4A, unsupervised analysis using the X-Shift clustering algorithm detected 10 clusters which were identified based on their expression of cell population markers (S6A and S6B Fig). Six of these clusters displayed APC features, characterized by their expression of MHC class I and class II molecules, as well as activation costimulatory markers such as CD80, CD86 [38,39], and CD24 [40] (Fig 4B). They were designated as dendritic cells subset 1 (cluster 2), B cells (cluster 3), neutrophils (cluster 4), dendritic cells subset 2 (cluster 5), monocytes (cluster 7), and NKT cells (cluster 10).

To assess the impact of Treg cell depletion on the frequency and phenotype of APCs and innate immune cells, we conducted a supervised analysis of our cytometric data. None of the six identified clusters showed significant changes in frequency between non-infected samples, Treg-depleted and non-depleted infected samples at the studied time point (S6C Fig). However, we found that Treg cell depletion led to increased frequencies of CD86+ cells within the large cluster of B cells (C3) and, surprisingly, in NKT cells (C10), which showed an unexpected great proportion of cells expressing this co-stimulatory molecule (Fig 4C and 4D). A similar trend was also observed in one DC cluster (C2). Interestingly, DT-treated mice showed increased frequency of CD86+ cells compared to non-infected controls in the clusters of B cells (C3), DC subset 2 (C5) and NKT cells (C10). No significant differences were observed in the frequency of positive cells for the remaining markers studied across the experimental groups within any of the identified clusters (S6D Fig). Taken together, these data point to subtle effects of Treg cells on APC and innate immune cell populations at early stages following *T. cruzi* infection.

A previous study has demonstrated that CD4+ T cell help is required to mount a full-sized TSKB20-specific CD8+ T cell response to *T. cruzi* infection [5]. Since Foxp3- CD4+ T (Tconv) cells are potential targets for Treg cell suppression [41–43], we investigated the impact of Treg cell depletion on this effector cell population. At day 20 pi, animals that received PBS or DT on days 5 and 6 pi showed similar frequencies of Tconv cells in blood, spleen, and liver (S7A Fig), with a tendency for higher counts in the spleen and significantly reduced numbers in the liver (S7B Fig). In contrast, when analyses were performed at day 11 pi, we observed a significant increase in the percentage of Tconv cells in the tissues from Treg-depleted mice, along with

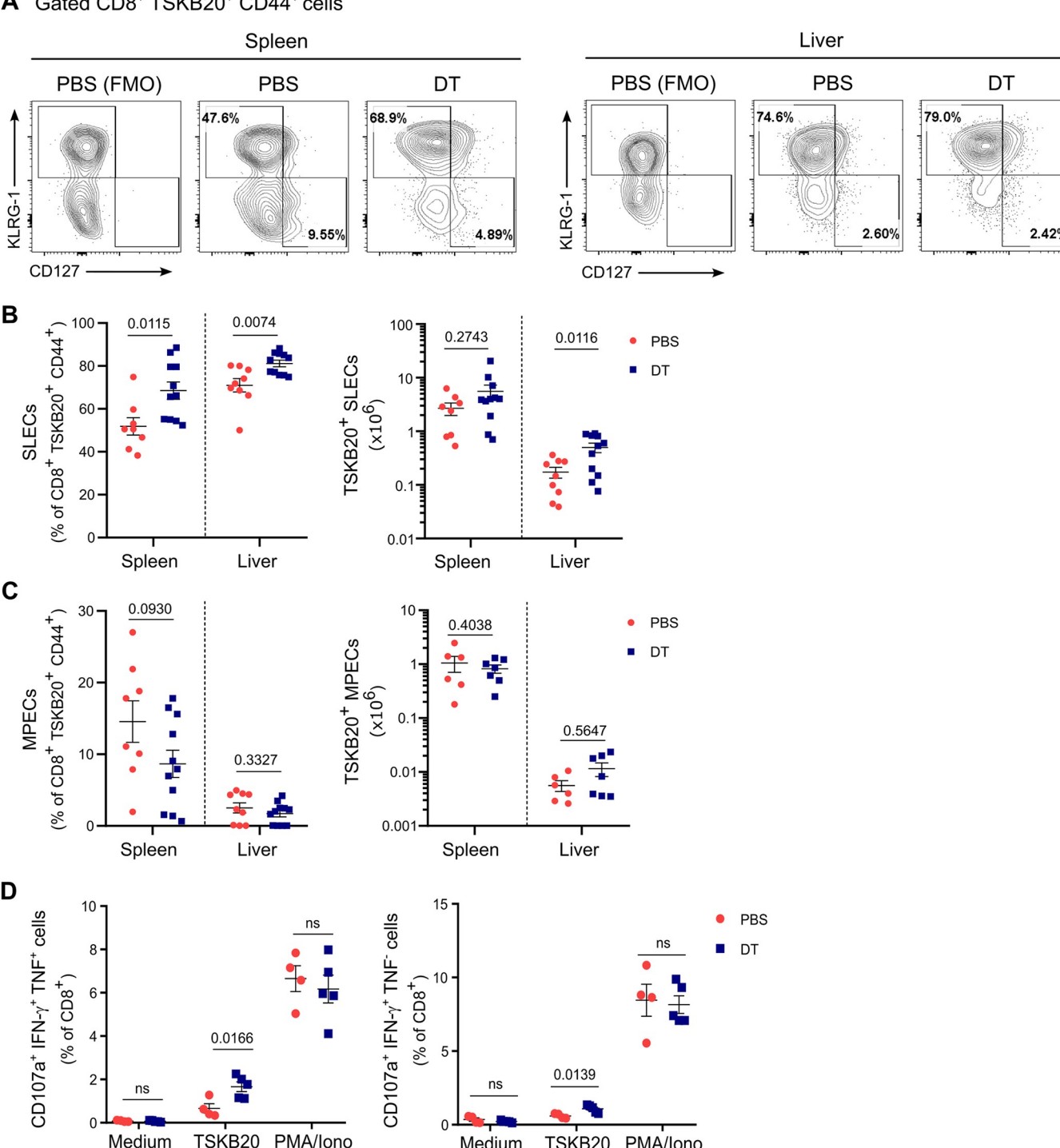

**Fig 3. Early Treg cell depletion promotes parasite-specific CD8+ T cell differentiation into SLEC during *T. cruzi* infection. A)** Representative flow cytometry plots showing KLRG-1+ CD127- (SLEC) and KLRG-1- CD127+ (MPEC) subsets within CD44+ gated TSKB20-specific CD8+ T cells in the indicated organs obtained from PBS or DT-treated DEREG mice at day 20 pi. **B-C)** Frequencies (left) and absolute numbers (right) of SLEC (B) and MPEC (C) subsets within CD44+ gated TSKB20-specific CD8+ T cells of mice in (A). Data were collected from 2 independent experiments. **D)** Percentages of CD8+ T cells that produce IFN-γ and exhibit CD107a mobilization together (left) or not (right) with TNF production in the spleen of PBS or DT-treated DEREG mice at day 21 pi. Medium condition was used as a negative control, while PMA/Ionomycin (PMA/Iono) was used as a positive control for polyclonal CD8+ T cell stimulation. Similar results were obtained in 2 independent experiments. All data are presented as mean ± SEM. In (B), (C) and (D) each symbol represents one individual mouse. Statistical significance was determined by Unpaired t test or Mann Whitney test, according to data distribution. P values for pairwise comparisons are indicated in the graphs.

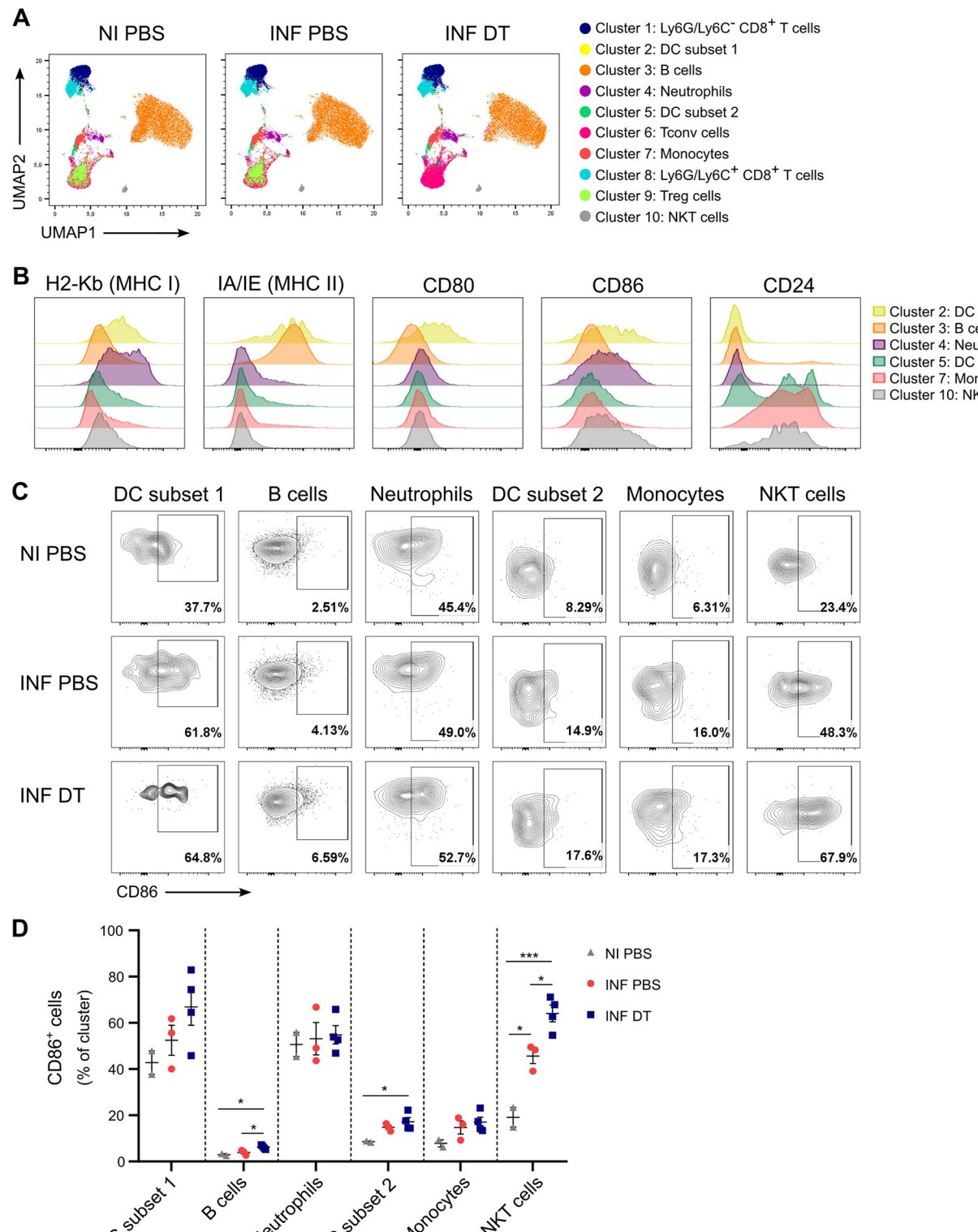

**Fig 4. Treg cell depletion induces modest effects on APC populations and innate cells. A)** UMAP visualization of flow cytometry data from the spleen of PBS or DT-treated DEREG mice at day 7 pi and PBS-treated non-infected controls. **B)** Histograms showing the expression of different APC and innate cell activation markers in selected cell clusters. Samples from the three experimental groups (NI PBS, INF PBS and INF DT) were pooled together. **C)** Representative flow cytometry plots showing CD86+ cells in the indicated cell clusters as defined in (A). **D)** Frequency of CD86+ cells in the indicated cell clusters as defined in (A). Data are presented as mean ± SEM. Each symbol represents one individual mouse. Statistical significance

was determined by one-way ANOVA followed by Tukey's multiple comparison test. Similar results were obtained in 3 independent experiments. * P ≤ 0.05, *** P ≤ 0.001.

augmented absolute Tconv cell counts in the liver (Fig 5A and 5B). Further analysis of subset distribution based on differentiation status revealed that effector Tconv cells were increased in the spleen and liver of DT-injected mice, at the expense of a reduction in the frequency of

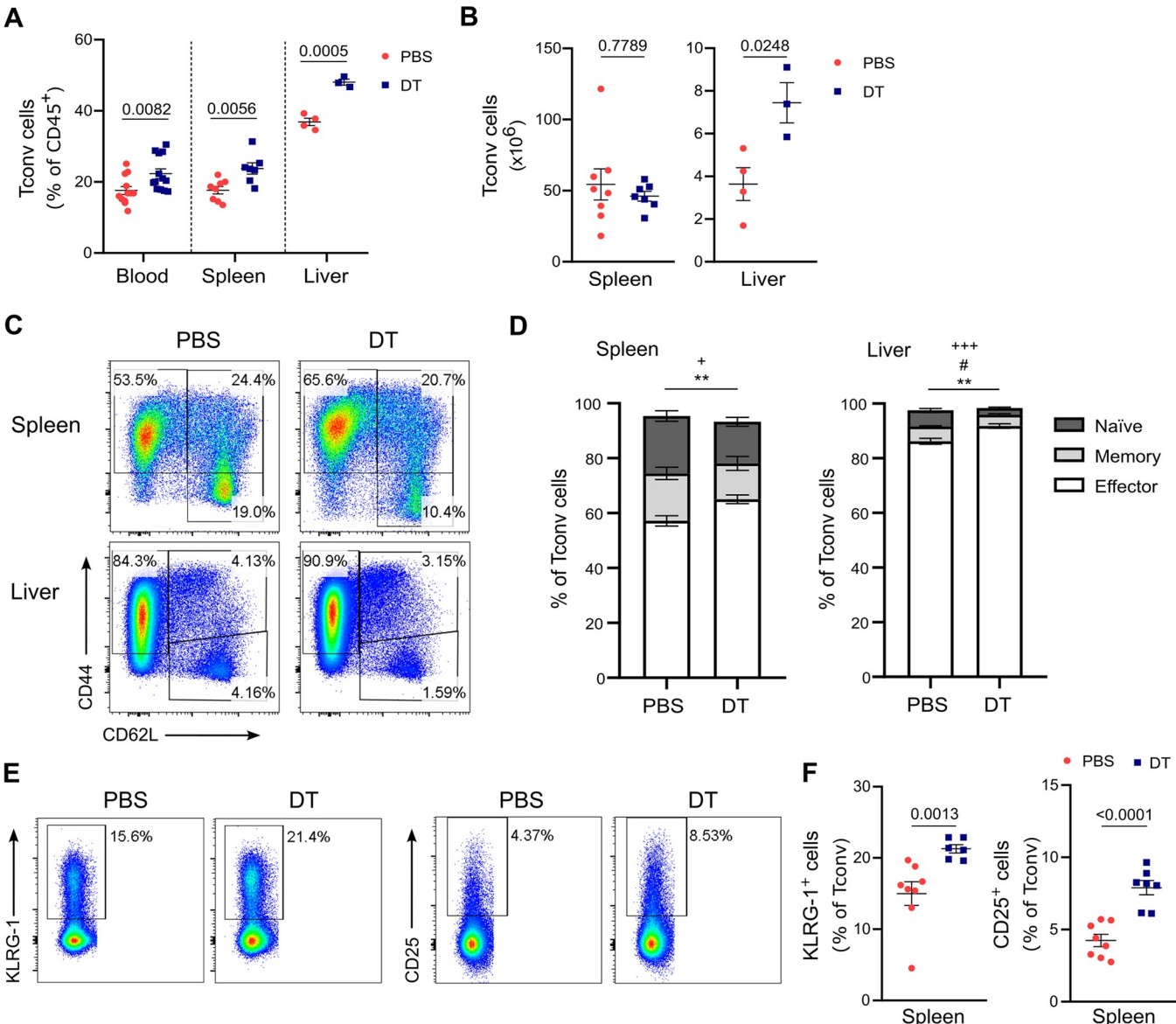

**Fig 5. Treg cell depletion promotes the expansion and activation of Tconv cells in *T. cruzi* target organs. A-B)** Frequencies (A) and absolute numbers (B) of Tconv cells in blood, spleen, and liver from PBS or DT-treated DEREG mice at day 11 pi. **C-D)** Representative flow cytometry plots (C) and frequency (D) of CD44- CD62L+ (naïve), CD44+ CD62L+ (memory) and CD44+ CD62L- (effector) Tconv cell subsets in the spleen and liver from mice in (A). **E-F)** Representative flow cytometry plots (E) and frequency (F) of KLRG-1+ and CD25+ Tconv cells in the spleen from mice in (A). All data are presented as mean ± SEM. Each symbol represents one individual mouse. Data in (A), (B), (D) and (F) were pooled from 1–2 independent experiments. Statistical significance was determined by Unpaired t test or Mann Whitney test, according to data distribution. P values for pairwise comparisons are indicated in the graphs. In (D), P values PBS vs DT: + P≤ 0.05 and +++ P ≤ 0.001, naïve Tconv cells; # P<0.05, memory Tconv cells; ** P ≤ 0.01, effector T conv cells.

naïve or naïve and memory Tconv cell subsets, respectively (Fig 5C and 5D). Moreover, we observed an increased frequency of Tconv cells expressing the T cell activation markers KLRG-1 and CD25 in the spleen, and KLRG-1 specifically in the liver, of Treg-depleted animals at day 11 pi (Figs 5E, 5F, and S7C). These results suggest that Treg cells influence helper responses by modulating Tconv cell activation state and numbers at early stages of *T. cruzi* infection coincident with the initiation of the parasite-specific CD8+ T cell response.

## TSKB20-specific CD8+ T cell suppression is mediated by CD39 expression on Treg cells

In order to gain mechanistic insight into how Treg cells impact on the magnitude of the parasite-specific CD8+ T cell response, we then focused on suppressive molecules upregulated by Treg cells after *T. cruzi* infection. It is long known that Treg cells constitutively express key molecules associated with their suppressive function, including CTLA-4 and CD25 in addition to Foxp3 [44]. In line with our previous findings that highlighted CD25, CTLA-4, and CD39 among the most upregulated markers in Treg cells during the peak of effector CD8+ T cell immunity [14], we sought to determine whether these molecules were also upregulated at earlier time points. Therefore, we analyzed their expression by flow cytometry at day 7 pi. Interestingly, while the frequency and absolute numbers of CD25+ and CTLA-4$^{hi}$ Treg cells showed no differences between infected and non-infected mice at this time point, there was a significant increase in the proportion and cell counts of Treg cells expressing high levels of CD39 in the spleen of *T. cruzi*-infected animals compared to controls (Fig 6A, 6B and 6C). Of note, an increased frequency and number of cells with high expression of CD39 were also observed in other cell subsets from infected mice, particularly non-CD4+ cells (S8A–S8C Fig). Additionally, both Tconv and non-CD4+ cells showed a tendency toward higher expression of this molecule, as indicated by the increased geometric mean of the CD39 staining (S8D Fig).

The upregulation of CD39 in the Treg cell subset (and in other immune cell populations) in the context of *T. cruzi* infection prompted us to investigate its potential role in the progression of this parasite infection. As a first approach, we utilized CD39 deficient mice that have a complete deletion of this molecule in all cell subsets. Remarkably, despite similar levels of blood parasitemia were observed between WT and CD39 KO infected mice (Fig 6D), mice deficient in CD39 exhibited an increased frequency of parasite-specific CD8+ T cells in both blood and spleen (Fig 6E and 6F). These findings support the notion that CD39-mediated pathways are involved in the regulation of CD8+ T cell immunity during *T. cruzi* infection.

To investigate whether ATP hydrolysis and ADO production may be the mechanisms underlying these effects, we quantified ATP and ADO in the plasma of WT and CD39 KO mice. As shown in Fig 6G, deficiency in CD39 KO results in a significant increase in the concentration of plasmatic ATP as determined at 18 dpi, which otherwise remains similar in infected WT mice in comparison to non-infected, WT and CD39 KO mice. No changes in plasmatic ADO levels were observed among non-infected and infected, WT and CD39 KO animals at the same time point. Remarkably, we determined that Treg cell depletion had no effect on the levels of ATP and ADO in plasma nor in the supernatants of spleen, liver, and heart, at least as determined at 21 dpi (Figs 6H and S8E).

Given the potential limited global impact of CD39$^{hi}$ Treg cells as a minor cell subset in the context of overall CD39 upregulation during *T. cruzi* infection, or the possibility that ATP levels are recovered by 21 dpi after early Treg cell depletion, the lack of effect of DT treatment on ATP levels does not preclude the possibility that CD39 expression may be relevant for Treg cell-mediated suppression of CD8+ T cell responses and parasite control. To specifically evaluate this, we conducted adoptive transfer experiments to assess whether the expression of CD39

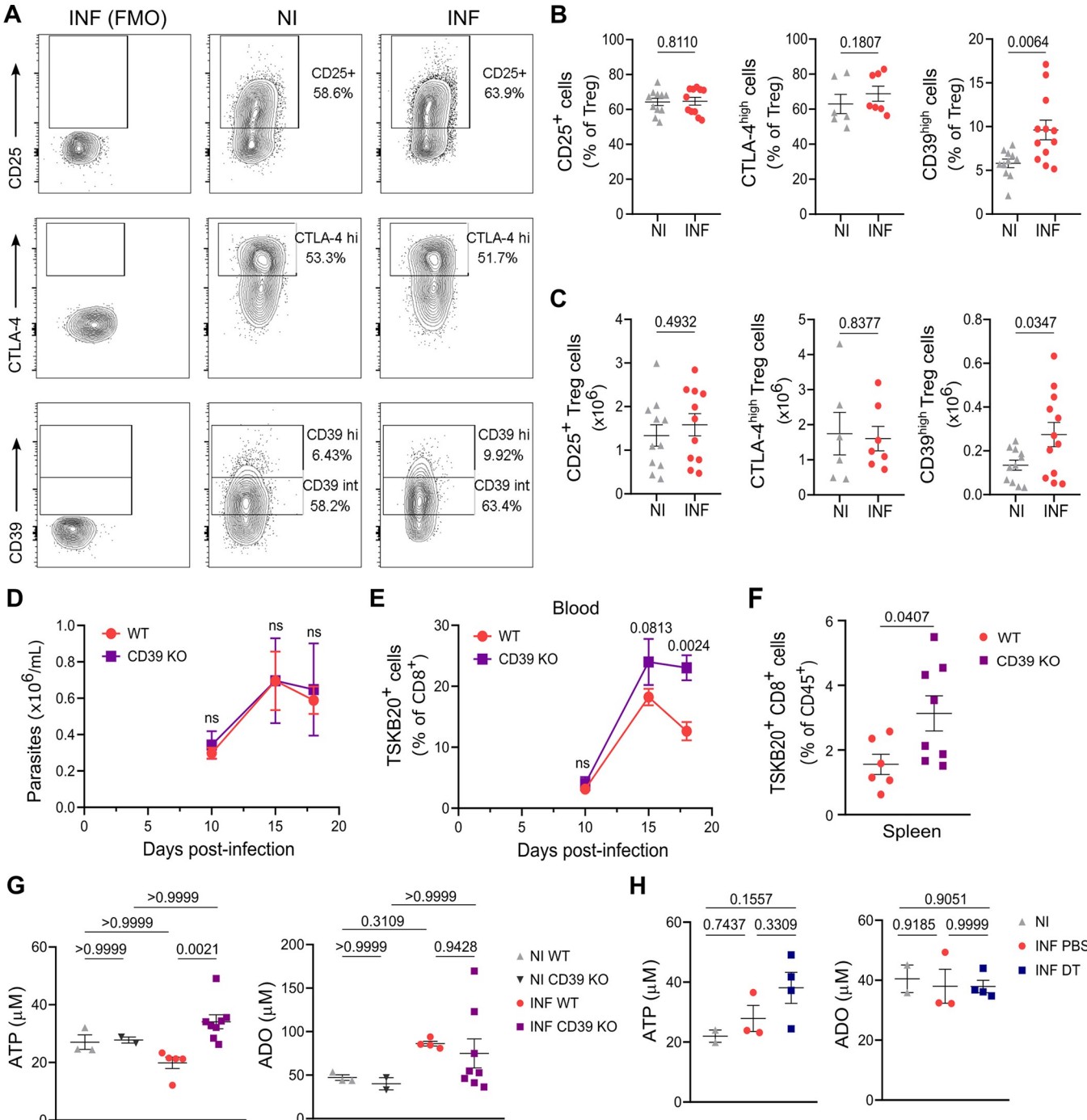

**Fig 6. CD39 is early upregulated during *T. cruzi* infection and regulates TSKB20-specific CD8+ T cell responses. A-C)** Representative flow cytometry plots (A), frequency (B) and absolute numbers (C) of Treg cells expressing CD25, CTLA-4 and CD39 in the spleen of DEREG mice at day 7 pi (INF) and non-infected controls (NI). Data were collected from 2–3 independent experiments. **D-F)** Parasite blood counts (D), and frequency of TSKB20+ CD8+ T cells in the blood (E) and spleen (F) of WT and CD39 KO INF mice at different dpi (D, E) and at day 18 pi (F). **G-H)** Concentration of ATP and Adenosine (ADO) quantified in the plasma of WT and CD39 KO, NI and INF (18 dpi) mice (G), or in the plasma of NI, PBS or DT-treated INF (20 dpi) mice (H). Data in (D-H) were collected from 1 experiment. All data are presented as mean ± SEM. In (B-C), and (F-H) each symbol represents one individual mouse. Statistical significance was determined by Unpaired t test or Mann Whitney test in (B-F), by Kruskal-Wallis test followed by Dunn's multiple comparisons test in (G) and by one-way ANOVA followed by Tukey's multiple comparisons test in (H). Values for pairwise comparisons are indicated in the graphs. ns = not significant.

by Treg cells is necessary for suppressing the TSKB20-specific CD8+ T cell response after *T. cruzi* infection. As illustrated in Fig 7A, DEREG mice were infected and Treg cell depletion was carried out as usual by DT administration at days 5 and 6 pi. The following day, a group of DT-treated mice received *in vitro* differentiated Treg (iTreg) cells from CD4+ splenocytes of non-infected WT or CD39 KO animals. As expected, Treg cell depletion led to an increased TSKB20-specific CD8+ T cell response at day 22 pi in the spleen and liver (Fig 7B and 7C). Notably, depleted animals that were injected with WT iTreg cells reversed the effect of DT treatment, as demonstrated by a reduction in the parasite-specific CD8+ T cell response to levels similar to those of PBS-injected (Treg-sufficient) controls. In contrast, injection of CD39 KO iTreg cells in depleted mice resulted in a robust *T. cruzi*-specific CD8+ T cell response comparable to or higher than DT-treated animals, and notably higher than Treg-depleted mice that received WT iTreg cells, in both the spleen and liver (Fig 7B and 7C). Consistent with our previous findings, the majority of TSKB20-specific CD8+ T cells displayed a SLEC phenotype at day 22 pi in the spleen and liver of Treg-depleted mice (Fig 7D). Interestingly, the relative and absolute numbers of SLEC TSKB20-specific CD8+ T cells significantly decreased when Treg-depleted animals received WT iTreg cells, but not when they were transferred with CD39 KO iTreg cells. Importantly, while *T. cruzi* numbers in blood showed no difference (Fig 7E), there was a significant increase in parasite load in the spleen and liver of Treg-depleted mice transferred with WT iTreg cells compared to those receiving CD39 KO iTreg cells (Fig 7F).

Collectively, these findings provide compelling evidence for the involvement of CD39-expressing Treg cells in modulating the magnitude of the effector *T. cruzi*-specific CD8+ T cell response and the control of parasite replication during the acute phase of infection. Importantly, this regulatory function was not compensated by other suppressive molecules expressed in Foxp3+ regulatory T cells, nor by CD39 expressed in other cell subsets in the context of *T. cruzi* infection.

## Discussion

Regulatory T cells play a pivotal role in infections by maintaining an effective balance between efficient immune responses for pathogen clearance and excessive immune reactivity that can lead to tissue damage. While the accumulation of Treg cells in the chronic phase has been well-documented for various infections, the contribution to immunity and disease outcome during the acute phase appears more highly context-dependent [45].

In this study, we demonstrate the crucial role of Treg cells in modulating the anti-parasite immune response and controlling pathogen replication during the acute phase of experimental *T. cruzi* infection. This work complements our previous study on the Treg cell response [14]. Here, we have elucidated the timing and mechanisms through which activated Treg cells, despite a relative decrease in numbers, exert a suppressive function on effector CD8+ T cell immunity during acute *T. cruzi* infection. These results differ from previous observations where discrepancies were reported regarding the role of Treg cells in this parasitic infection, as some studies described beneficial impacts [29,46], while others suggested limited [47,48] or deleterious effects [49]. However, none of these studies have thoroughly investigated the phenotypical and functional characteristics of the Treg cell response, and more importantly, all of them targeted these cells by non-specific approaches.

Taking advantage of a diphtheria toxin (DT) model for Treg cell depletion, we found that removal of endogenous Treg cells shortly after infection, but not at later stages, improved parasite control in blood as well as in certain *T. cruzi* target tissues. The differential impact of Treg cell depletion on parasite loads in tissues may reflect specific patterns associated with parasite

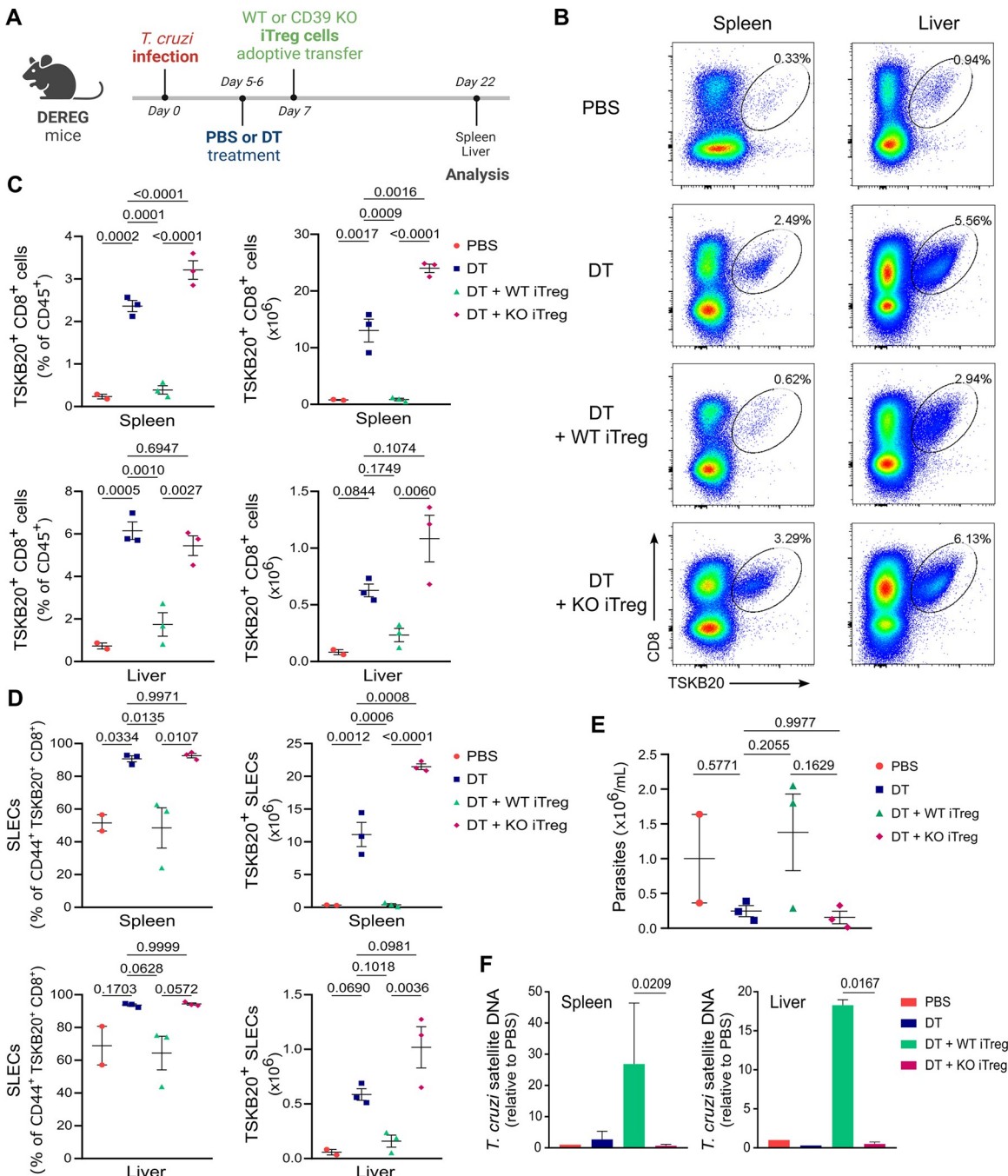

**Fig 7. CD39 expression on Treg cells mediates TSKB20-specific CD8+ T cell suppression and parasite outgrowth. A)** Overview of iTreg cell adoptive transfer experimental design (created with BioRender.com). **B-C)** Representative flow cytometry plots (B) and frequency and absolute numbers (C) of TSKB20+ CD8+ T cells in the spleen and liver of mice treated and analyzed as indicated in (A). **D)** Frequencies and absolute numbers of the SLEC subset within CD44+ gated TSKB20-specific CD8+ T cells from mice treated as indicated in (A). **E-F)** Parasite counts in blood (E) and parasite load in the spleen and liver (F) from mice treated as indicated in (A). Results similar to (C-F) were obtained in 2 independent experiments. All data are presented as mean ± SEM. In (C-E) each symbol represents one individual mouse. Statistical significance was determined by one-way ANOVA followed by Tukey's multiple comparisons test in (C-E), and by paired t test in (F). Values for pairwise comparisons are indicated in the graphs.

tropism [50,51], as well as particular host-parasite dynamics at local level [52], and may require systematic evaluation in a variety of relevant tissues including skeletal muscle, visceral adipose tissue, and colon among others. Nonetheless, we propose that the effect of Treg cell depletion in parasite control is secondary to an overall enhancement of TSKB20-specific CD8+ T cell immunity without causing significant tissue damage at the peak of the CD8+ T cell response. These results align with a previous study that reported increased numbers of CD8+ T cells restricted to the VNHRFTLV epitope of Amastigote Surface Protein-2 from *T. cruzi* when Treg cells were ablated [15]. Overall, these observations highlight the role of Treg cells in suppressing optimal CD8+ T cell-mediated immunity against *T. cruzi* and indicate that Treg cells play relevant functions in this parasitic infection.

Following activation by antigen encounter, CD8+ T cells undergo expansion and differentiation into effector cells, which can be categorized into two distinct subsets: highly functional short-lived effector cells (SLEC) that will undergo apoptosis after antigen clearance, and memory-precursors (MPEC) that are able to continue the differentiation program into long-lived memory cells. In the current study, we provide evidence that Treg cells limit the magnitude of parasite-specific CD8+ T immunity, by preferentially reducing the SLEC subset while having no impact on the generation of parasite-specific MPECs. Our results contrast with previous observations where Treg cells either promote [53–56] or restrict [57] the development of the CD8+ T cell memory response in viral or bacterial infections and immunization models. These discrepancies might be attributed to variations in the inflammatory milieu of each setting, considering the essential role of cytokines in directing CD8+ T cell differentiation programs, including IL-2 and IL-10 that mediate Treg cell function in those studies. Additionally, we demonstrated that Treg cells seem to exert minimal influence on the acquisition of other phenotypic attributes by CD8+ T cell responses, such as the expression of activation and exhaustion markers or functional molecules. Altogether, these data reinforce the notion that the impact of Treg cell regulation is quantitative rather than qualitative within CD8+ T cell immunity against *T. cruzi*.

Multiple regulatory mechanisms have been described for Treg cells, such as the secretion of suppressive cytokines and cytolytic molecules, expression of inhibitory receptors and metabolic disruption of effector cells. However, there is still limited understanding of the precise mechanisms employed by Treg cells to mediate their suppressive activity in different inflammatory settings. Acquiring this knowledge will enable the modulation of inflammation outcomes associated with infections by therapeutic approaches targeting Treg cells. In recent years, purinergic suppressive pathways have gained significance. Treg cells have been shown to control inflammation in experimental models and in clinical studies, mediated, at least in part, through an adenosine-dependent manner via the regulated expression of CD39 and CD73 [22,58–62]. In particular, it has been reported that human and mouse Treg cells expressing CD39 suppress Tconv and CD8+ T cell proliferation or cytokine production more efficiently *in vitro*, when compared to Treg cells lacking CD39 expression [63,64]. Additionally, CD39-expressing Treg cells exhibit stronger *in vitro* suppressive effects than CD39- Treg cells or upon CD39 inhibition [65–67].

Previously, we reported that Treg cells upregulate CD39 expression along with a wide range of suppressive markers and regulatory molecules during the acute phase of *T. cruzi* infection [14]. Variations in CD39 expression within Treg cell subsets, particularly in human samples, have been documented but not further functionally evaluated [68,69]. We have also demonstrated that, in the case of CD8+ T cells activated in the context of cancer, high versus intermediate expression of CD39 defines subsets with distinct phenotypic and functional characteristics. CD39hi cells display higher activation, increased expression of inhibitory receptors, and enhanced ATP hydrolysis capacity [24]. In our study, we observed an early

upregulation of CD39 expression soon after infection, with CD39[hi] cells arising in the Treg cell pool as well as in other cell subsets. Building on these findings and the reported role of CD39 as part of an immunosuppressive pathway [21], we utilized CD39 KO mice to effectively demonstrate its involvement in regulating parasite-specific CD8+ T cell responses during *T. cruzi* infection.

As anticipated, CD39 KO mice exhibited increased ATP levels and conserved ADO levels at the peak of infection, potentially accounting for the observed enhancement in CD8+ T cell immunity. In contrast, Treg cell-depleted mice showed no alterations in ATP or ADO levels. We propose two plausible explanations for this unexpected result. Firstly, ATP levels may have normalized by the peak of infection due to early Treg cell depletion. Alternatively, the elimination of a minor population such as CD39+ Treg cells may have limited systemic impact on ATP hydrolysis, given the numerous cell populations upregulating this molecule during *T. cruzi* infection. A similar intricate scenario is being discussed in the context of tumor microenvironments [70]. Nonetheless, these findings do not rule out the potential involvement of CD39 in Treg cell-mediated immunosuppression of parasite-specific CD8+ T cell responses observed in our *T. cruzi* setting. This hypothesis is supported by the notion that Treg cells operate in micro-domains where they closely interact with the cells they regulate [71,72]. In fact, our adoptive transfer experiments definitively demonstrated that CD39 expression on Treg cells is necessary for controlling parasite-specific CD8+ T cell responses, independent of other regulatory molecules expressed on the Treg cells and despite CD39 expression in other cell populations.

To our knowledge, this is the first description of *in vivo* T cell suppression mediated by CD39+ Treg cells in the context of acute *T. cruzi* infection. Further research is needed to understand the precise mechanism involved, which may include an early and transient modification of ATP/ADO metabolism or a CD39-driven acquisition of a more robust regulatory program in Treg cells [68]. Additionally, investigating the impact of early Treg cell depletion or targeting CD39 during the chronic phase of *T. cruzi* infection may provide further evidence for its use as a complementary approach to parasiticidal drugs, aiming to enhance effector anti-parasite responses. Remarkably, a recent report showed that patients with chronic chagasic cardiomyopathy exhibit expansion of activated effector CD4+ T cell subsets, which correlate with a decreased frequency of CD39+ Treg cells. These data suggest that suppressive mechanisms also operate in the context of the human infection [73].

Treg cell-mediated suppression of effector CD8+ T cell responses can occur through direct contact/proximity between the cells [74,75], indicating a direct suppressive effect, or through the modulation of other immune cell populations that are important for CD8+ T cell functions [76], suggesting an indirect regulatory mechanism. In this study, we provide clear evidence that Treg cells not only impact CD8+ T cell immunity but also restrain Tconv cell responses during acute *T. cruzi* infection. Our observation that Treg cell depletion affects Tconv cell numbers and their activation phenotype prior to the regulation of parasite-specific CD8+ T cells suggests an indirect effect of Treg cells on parasite-specific CD8+ T cells through the modulation of Tconv cells. However, we cannot rule out the possibility of direct interactions between Treg cells and CD8+ T cells. CD4+ T cells have long been recognized for their role in providing help for CD8+ T cell priming, particularly in the generation of memory CD8+ T cell pools [77]. The "licensing" model has been proposed as the main mechanism for CD4+ T cell help delivery, in which these cells are necessary to enhance the antigen-presenting and co-stimulatory capacities of DCs to induce robust effector CD8+ T cell responses [78]. In agreement with our results, a study found that activation of CD4+ T helper cells preceded that of CD8+ T cells and that this process involved the action of different DC subsets in the context of HSV infection [79]. In our system, APCs were only mildly affected by Treg cell depletion.

Nevertheless, it would be interesting to evaluate the impact of early Treg cell elimination on APCs at later time points coinciding with the effect on the Tconv cell response in order to establish if a Treg-Tconv-APC axis is involved in the modulation of parasite-specific CD8+ T cells during *T. cruzi* infection. Furthermore, it has been reported that adenosine can downregulate CD86 expression on DCs [80,81]. Whether CD39 activity on Treg cells is responsible for the modulation of Tconv cell and APC responses needs to be further addressed in this parasitic infection.

NKT cells are a group of innate-like T cells that recognize lipids in the context of CD1d molecules. By responding very rapidly to TCR and/or cytokines they can link innate and adaptive immune responses and are important players in infectious diseases [82]. The contribution of NKT cells to protection against *T. cruzi* remains controversial in the context of the experimental infection [83–88], while few studies have focused on this cell population in human Chagas disease. For instance, established chronic infection in asymptomatic patients can be accompanied by increased peripheral blood frequencies of both NKT cells and regulatory CD25+ T cells, and these values were inversely correlated to numbers of activated CD8+ T cells [19,89]. In this study, we found that *T. cruzi* infection augmented the frequency of NKT cells expressing CD86 and that this effect was even increased after Treg cell depletion, potentially linked to CD8+ T cell priming during the acute phase. The implications of CD86 expression by NKT cells, the interaction between NKT cells and Treg cells, and their functional properties in the context of *T. cruzi* infection deserve further investigation.

In summary, the current study supports our previous hypothesis that weakened Treg cell responses during the acute phase of *T. cruzi* infection have beneficial elements for the host. This mechanism allows the generation of robust anti-parasite responses that prevent pathogen outgrowth without enhancing tissue damage. Further research is required to evaluate any sequelae of early Treg cell depletion. One potential outcome is that improved control of the parasite burden in the acute phase limits persistence of *T. cruzi* in target tissues during the chronic stage. Conversely, the reinvigorated effector response may prevail, resulting in immunopathology. Additionally, the restriction of Treg cells during the chronic phase should also be carefully examined, as this may trigger harmful inflammatory responses by leveraging chronic effector responses. In fact, previous studies of the human infection have linked impaired Treg cell immunity to more severe clinical forms of Chagas disease [16,19,90]. Therefore, more detailed investigations are necessary to fully understand the role of Treg cells and CD39 expression in the transition from acute to chronic *T. cruzi* infection and the development of disease.

## Materials and methods

### Ethics statement

Mouse handling followed international ethical guidelines. All experimental procedures were conducted in compliance with the ethical standards set by the Institutional Animal Care and Use Committee of Facultad de Ciencias Químicas, Universidad Nacional de Córdoba (FCQ-UNC), and were approved under protocol numbers RD-731-2018 and RD-2134-2022.

### Mice

Male and female mice aged 8 to 12 weeks were used for experiments. C57BL/6 and BALB/c wild type mice were obtained from the School of Veterinary, La Plata National University (La Plata, Argentina). DEREG (C57BL/6-Tg(Foxp3-DTR/EGFP)23.2Spar/Mmjax) reporter mice were purchased from The Jackson Laboratories (USA). CD39 deficient mice were developed and provided by Dr. Simon Robson [91]. All mouse strains were bred and housed at the animal

facility of the FCQ-UNC. Mice were maintained on a 12-hour light/12-hour dark cycle with food and water *ad libitum*. Mice from different experimental groups were co-housed in the same cages. DEREG mice were bred as heterozygotes on a C57BL/6 background, and F1 generation was routinely tested for GFP expression.

### Parasites and experimental infection

Bloodstream trypomastigotes of the Tulahuén strain of *T. cruzi* were maintained in male BALB/c mice by serial passages every 10–11 days. For experimental infection, mice were intraperitoneally inoculated with 0.2mL PBS containing $5 \times 10^3$ trypomastigotes obtained by diluting the blood from parasite passages (typically diluted around 400X). All infections were performed at similar hours of the day.

### Treg cell depletion

Mice were randomly divided into Treg cell depleted and control groups. For Treg cell depletion, DEREG mice were injected intraperitoneally with 25 ng of Diphtheria Toxin (DT; Calbiochem) per gram of body weight (25 ng/g) diluted in PBS. DT was administered in two consecutive days at the indicated time points. Control non-depleted mice received PBS injections. The following day, Treg cell ablation was confirmed in blood samples by flow cytometry.

### Parasite quantification

Parasitemia was assessed by counting the number of viable trypomastigotes in blood after lysis with a 0.87% ammonium chloride buffer. Abundance of *T. cruzi* satellite DNA in tissues was used to determine parasite burden. Genomic DNA was purified from 50 μg of tissue (heart, liver, and spleen) with TRIzol Reagent (Life Technologies) following manufacturer's instructions. DNA samples from the same tissue and experimental group were pooled in each experiment. Satellite DNA from *T. cruzi* (GenBank AY520036) was quantified by real time PCR using specific Custom Taqman Gene Expression Assay (Applied Biosystems). Primers and probes sequences were previously described by Piron *et al.* [92]. The samples were subjected to 45 PCR cycles in a thermocycler StepOnePlus Real-Time PCR System (Applied Biosystems). Abundance of satellite DNA from *T. cruzi* was normalized to the abundance of GAPDH (Taqman Rodent GAPDH Control Reagent, Applied Biosystems), quantified through the comparative ΔΔCT method and expressed as arbitrary units, as previously reported [14,28,33].

### Cell preparation

Blood was collected via cardiac puncture using heparin as anticoagulant. Spleens and livers were obtained and mechanically disaggregated through a tissue strainer to obtain cell suspensions in PBS 2% FBS. Liver infiltrating leukocytes were isolated by centrifugation at 600g for 25 minutes using a 35% and 70% bilayer Percoll (GE Healthcare) gradient. Erythrocytes were lysed for 3 min using an ammonium chloride-potassium phosphate buffer (ACK Lysing Buffer, Gibco). Cell numbers were counted in Turk's solution using a Neubauer chamber and used to calculate the number of specific subsets shown in several figures.

### Biochemical determinations

Blood samples were centrifuged at 3000 rpm for 8 min and plasma was collected. Quantification of GOT, GPT, LDH and CPK activities was performed by UV kinetic method, CPK-MB activity by enzymatic method, and glucose concentration by enzymatic/colorimetric method at Biocon Laboratory (Córdoba, Argentina) using a Dimension RXL Siemens analyzer.

## Flow cytometry

For surface staining, cell suspensions were incubated with fluorochrome labeled-antibodies together with LIVE/DEAD Fixable Cell Dead Stain (ThermoFisher) in PBS 2% FBS for 20 min at 4°C. To identify *T. cruzi* specific CD8+ T cells, cell suspensions were incubated with an H-2Kb *T. cruzi* trans-sialidase amino acids 569–576 ANYKFTLV (TSKB20) APC- or BV421-labeled Tetramer (NIH Tetramer Core Facility) for 20 min at 4°C, in addition to the surface staining antibodies. To detect NKT cells, cell suspensions were incubated with PBS-57, an analogue of α-galactosylceramide (α-Gal-Cer) [93], complexed to CD1d APC-labeled Tetramers (NIH Tetramer Core Facility). Blood samples were directly incubated with the specified antibodies, and erythrocytes were lysed with a 0.87% NH4Cl buffer prior to acquisition.

For the detection of transcription factors, cells were initially stained for surface markers, washed, fixed, permeabilized and stained with Foxp3/Transcription Factor Staining Buffers (eBioscience) according to eBioscience One-step protocol for intracellular (nuclear) proteins.

For the UMAP analysis, the cytometry panel included the following anti-mouse monoclonal antibodies: PerCP-eFluor 710 anti-CD80 (B7-1) clone 16-10A1 (eBioscience), Super Bright 436 anti-MHC Class I (H-2kb) clone AF6-88.5.5.3 (eBioscience), Super Bright 600 anti-MHC Class II (I-A/I-E) clone M5/114.15.2 (eBioscience), Super Bright 645 anti-CD11b clone M1/70 (eBioscience), Super Bright 702 anti-Ly-6G/Ly-6C clone RB6-8C5 (eBioscience), Brilliant Violet 785 anti-CD86 clone GL-1 (Biolegend), Alexa Fluor 700 anti-CD45 clone 30-F11 (eBioscience), APC/Cyanine7 anti-F4/80 clone BM8 (Biolegend), PE anti-CD3e clone 145-2C11 (eBioscience), PE-eFluor 610 anti-CD24 clone M1/69 (eBioscience), PE-Cyanine5 anti-CD19 clone eBio1D3 (1D3) (eBioscience), PE-Cyanine5.5 anti-CD8a clone 53–6.7 (eBioscience), PE-Cyanine7 anti-CD11c clone N418 (eBioscience). The staining also included Tetramers to identify NKT cells and the LIVE/DEAD Fixable Aqua Dead Cell Stain Kit, for 405 nm excitation (Invitrogen).

The detailed list of antibodies used in all the experiments can be found in S1 Table. All samples were acquired using a FACSCanto II (BD Biosciences) or a LSRFortessa X-20 (BD Biosciences) flow cytometer, and data were analyzed with FlowJo software versions X.0.7 and 10.8.1. During the analysis, gating coordinates were established using negative controls for population markers. For activation markers, regulatory molecules, or continuously expressed molecules, FMO controls were employed to set gating parameters.

## Determination of CD8+ T Cell effector function *in vitro*

Spleen cell suspensions were cultured for 5h in RPMI 1640 medium (Gibco) supplemented with 10% heat-inactivated FBS (Gibco), 2 mM glutamine (Gibco), 55 μM 2-ME (Gibco), and 40 μg/ml gentamicin. Cells were stimulated with 2.5 μM TSKB20 (ANYKFTLV) peptide (Genscript Inc.) or 50 ng/mL PMA plus 1 μg/mL ionomycin (Sigma-Aldrich) in the presence of Monensin and Brefeldin A (eBioscience). Culture medium was used as negative control. Anti-CD107a was included during the culture period. After surface staining, cells were fixed and permeabilized using Intracellular Fixation & Permeabilization Buffer Set (eBioscience) following manufacturer's instructions. Stained cells were acquired on a FACSCanto II (BD Biosciences) or a LSRFortessa X-20 (BD Biosciences) flow cytometer as before. Antibodies specifications are detailed in S1 Table.

## Determination of DT toxicity on parasites *in vitro*

Blood-derived *T. cruzi* parasites were used to infect monolayers of Vero cells. After 7 days of culture, trypomastigotes were collected from supernatants by centrifugation at 4000 rpm for 20 min and resuspended in supplemented RPMI medium, as previously described [94].

Subsequently, $3 \times 10^5$ trypomastigotes were incubated in the presence of DT at the final concentrations of 1.3, 2.6, 5.2, 10.4 and 20.8 nM, using the indicated medium in 96-well cell culture plates. PBS served as the vehicle negative control, while Benznidazole at final concentration of 25 μM was used as a positive control for parasite killing. After 24h, parasites were counted using a Neubauer chamber.

## Adoptive cell transfer

CD4+ T cells were isolated from pooled splenic cell suspensions by magnetic negative selection using EasySep Mouse CD4+ T Cell Isolation Kit (StemCell Technologies) according to the manufacturer's protocol. The isolated cells were further purified to obtain naïve CD4+ T cells (CD4+ CD25- CD44-) by cell sorting using a FACSAria II (BD Biosciences) instrument (see S1 Table for antibodies specifications). The sorted naïve CD4+ T cells were then incubated with a Treg cell differentiation cocktail in 96-well cell culture plates. For this, $2 \times 10^5$ cells were stimulated with plate-bound anti-CD3 and anti-CD28 antibodies (eBioscience) at concentrations of 2 and 1 μg/mL, respectively, in the presence of 20 ng/mL rmIL-2 (Biolegend), 5 ng/mL m/h rTGF-β (eBioscience) and 13.3 nM *all trans*-Retinoic Acid (Sigma) diluted in supplemented RPMI 1640 medium (Gibco). On day 4, cells were harvested, and a viability higher than 80% as well as a Foxp3 expression higher than 90% were confirmed by flow cytometry. One million *in vitro* differentiated Treg cells were subsequently injected intravenously into the retro-orbital sinus of DEREG recipient mice infected with *T. cruzi* and previously treated with DT. Non-transferred and PBS-treated mice were included as controls.

## ADO and ATP determinations

The concentrations of adenosine (ADO) and adenosine triphosphate (ATP) were determined in methanol-deproteinized plasma and tissue supernatants by high-performance liquid chromatography (HPLC; Agilent 1200, Agilent Technologies, Wilmington, DE, USA), equipped with an isocratic pump and a UV detector. Briefly, tissue supernatants were obtained by rinsing a fraction of the spleen and liver or the whole heart in PBS, followed by 1h incubation at 37°C in PBS plus gentamicin at a standardized volume (μL) calculated as tissue weight (mg)/0.3. Data acquisition and processing was performed using Agilent BootP software. Chromatographic separations were achieved by use of a Phenomenex C18 reverse phase column (150×4.6 mm, 5 μm particle size). The mobile phase consisted of 0.4% phosphoric acid:methanol (95:5, v/v), and the flow rate was 0.8 mL/min in isocratic mode. The injection volume was 20 μL and detection was at 257 nm. ADO and ATP peak areas were validated through the co-injection of standards (ADO and ATP, Sigma-Aldrich) together with the analyzed samples. Quantitative analysis was conducted by correlating the peak areas with the regression line of calibration curves, which were prepared over a range of 0.1 to 200 μM by diluting ADO and ATP standards with PBS.

## Statistics

Descriptive statistics were calculated for each experimental group. The normality of data distribution was assessed using Shapiro-Wilk normality test. Statistical significance of mean value comparisons was determined using two-tailed t-test or One-way ANOVA for normally distributed data, and two-tailed Mann Whitney test or Kruskal-Wallis test for non-normally distributed data, as appropriate. P values ≤ 0.05 were considered statistically significant. Outliers were identified using the ROUT method. GraphPad Prism 9.0 software was used for statistical analyses and graph creation. Data are presented as mean ± SEM. The majority of the data presented were collected from different independent experiments, as the calculated sample size

required to achieve statistical significance was divided across 1–3 independent assays. In other experiments, presented data are from a representative experiment, which indicates that the same statistically significant differences were reproduced across independent experiments. Sample size for each experiment is indicated in the figure legends, while the number of animals of each experimental group is shown in the scatter dot plots, unless stated otherwise.

### AI Language model assistance

We used ChatGPT (developed by OpenAI) to assist in refining the written content of this study. ChatGPT provided suggestions and corrections based on the input provided by the user, enhancing the clarity and grammar of the text. ChatGPT output was critically revised by the user to ensure it conveys the desired message.

## Supporting information

**S1 Fig. Impact of DT injection on the frequency of Treg cells and total CD8+ T cells during *T. cruzi* infection. A-B)** Kinetics analysis of Treg cells frequencies within gated CD4+ cells determined in blood (A), spleen and liver (B) from PBS or DT-treated, NI or *T. cruzi* INF DEREG mice. **C-E)** Frequency (C, D) and absolute numbers (E) of total CD8+ T cells determined at different time points in blood (C) and at 20 dpi in spleen and liver (D, E) of PBS or DT-treated, NI and INF DEREG mice. Data are presented as mean ± SEM. Data were collected from 1–7 independent experiments according to the tissue and dpi in (A-C) and from 4–5 independent experiments in (D-E). In (D-E) each symbol represents one individual mouse. A total of 2–36 mice per group were included. In (A) and (C) n = 2–3 for NI groups, n = 12–27 at 10 dpi, n = 7–14 at 12 dpi, n = 16–36 at 20 dpi, n = 6–10 at 25 dpi, n = 12–17 at 33 dpi, n = 5–7 at 55 dpi. In (B) n = 10–12 at 7 dpi, n = 3–4 at 11 dpi, n = 12–14 at 20 dpi, n = 4–5 at 33 dpi. Statistical significance was determined by Unpaired t test or Mann Whitney test, according to data distribution. Statistical analysis in A-C represents pairwise comparisons between INF PBS and INF DT groups. * P ≤ 0.05, ** P ≤ 0.01, *** P ≤ 0.001, **** P ≤ 0.0001 and ns = not significant. P values for pairwise comparisons at day 20 pi are indicated in the graphs.
(PDF)

**S2 Fig. Control of DT treatment toxicity and evaluation of its impact on biochemical markers of damage associated with the progression of infection. A-B)** Parasite counts (A) and TSKB20-specific CD8+ T cell frequencies (B) in blood from PBS or DT-treated *T. cruzi* infected WT littermate mice at 20 dpi. Data were collected from 3 independent experiments and are presented as mean ± SEM. Statistical significance was determined by Mann Whitney test. **C)** Parasites numbers counted after 24 h of culture with PBS, increased doses of DT or benznidazole (BZ). Data are presented as mean ± SD of technical triplicates from 1 experiment. **D)** Treg cell depletion effect on tissue damage markers: activities of glutamate-oxalacetic transaminase (GOT), glutamate-pyruvate transaminase (GPT), lactate dehydrogenase (LDH), creatine phosphokinase (CPK), and creatine phosphokinase of muscle and brain (CPK MB), as well as Glucose concentration in plasma of PBS or DT-treated DEREG mice at days 20 and 33 pi. Data were collected from 6 independent experiments at 20 dpi (n = 20–26) and from 1 experiment at 33 dpi (n = 4–5). Data are presented as mean ± SEM. Statistical significance was determined by Unpaired t test for GOT, LDH, CPK, and CPK MB activities and Glucose concentration, and by Mann Whitney test for GPT activity, according to data distribution. P values for pairwise comparisons at day 20 pi are indicated in the graphs. ns = not significant.
(PDF)

**S3 Fig. Depletion of Treg cells on days 11 and 12 pi had no impact on parasitemia levels or parasite-specific CD8+ T cell numbers. A)** Experimental scheme for DT treatment (created with BioRender.com). **B-C)** Treg cell frequencies in blood at different dpi (B) and in spleen at day 21 pi (C) from *T. cruzi*-infected DEREG mice treated with PBS or DT on days 11 and 12 pi. **D)** Parasitemia levels from mice in (A). **E-F)** TSKB20-specific CD8+ T cell frequencies in blood at different dpi (E) and in spleen at day 21 pi (F) of mice in (A). All data are presented as mean ± SEM. Data were collected from 1–2 independent experiments. A total of 2–6 mice per group were included. Statistical significance was determined by Unpaired t test or Mann Whitney test, according to data distribution. P values for pairwise comparisons are indicated in the graphs. * P $\leq$ 0.05.
(PDF)

**S4 Fig. Effect of early Treg cell depletion on total and TSKB20-specific CD8+ T cell phenotype. A)** Gating strategy for evaluation of SLEC and MPEC subsets. **B)** Frequencies of SLEC (left) and MPEC (right) subsets within CD44+ gated CD8+ T cells from PBS or DT-treated DEREG mice at day 20 pi. Data were collected from 2 independent experiments and are presented as mean ± SEM. **C)** Comparison of the frequencies of cells expressing the indicated activation, exhaustion and functional markers in total and TSKB20-specific CD8+ T cells in the spleen and liver of PBS or DT-treated DEREG mice at day 20 pi. Data were collected from 1–4 independent experiments. Violin plots depict the distribution of frequency of 3–18 mice per group. Statistical significance was determined by Unpaired t test or Mann Whitney test, according to data distribution. P values for pairwise comparisons are indicated in the graphs.
(PDF)

**S5 Fig. Effect of early Treg cell depletion on CD8+ T cell function. A)** Gating strategy for assessing effector cytokine production and CD107a surface mobilization in gated CD8+ cells. **B)** Percentage of CD8+ T cells from the spleen of PBS or DT-treated DEREG mice at day 21 pi that exhibit different combinations of effector functions, including CD107a mobilization and/ or IFN-γ and/or TNF production upon 5h of the indicated stimulation. Medium condition was used as a negative control, while PMA/Ionomycin (PMA/Iono) was used as a positive control for polyclonal CD8+ T cell stimulation. Similar results were obtained in 2 independent experiments. All data are presented as mean ± SEM. Each symbol represents one individual mouse. Statistical significance was determined by Mann Whitney test.
(PDF)

**S6 Fig. Frequencies and expression of activation markers in APC and innate cell clusters. A)** UMAP visualization for splenocytes expression of the different population and activation markers used in the flow cytometry panel for APC and innate cells characterization. Samples from the three experimental groups (NI PBS, INF PBS and INF DT) are shown together. **B)** Heat map showing the expression level of each marker in the different clusters. **C)** Frequencies of selected clusters in total leukocytes (CD45+ cells) from the spleen of PBS or DT-treated DEREG mice at day 7 pi and non-infected controls. **D)** Frequency of cells expressing the indicated markers in the different clusters defined in Fig 4A. Clusters without positive cells for the corresponding marker were excluded from the analysis. All data are presented as mean ± SEM. Each symbol represents one individual mouse. Similar results were obtained in 3 independent experiments.
(PDF)

**S7 Fig. Effect of early Treg cell depletion on Tconv cell response. A-B)** Frequencies (A) and absolute numbers (B) of Tconv cells in blood, spleen and liver from PBS or DT-treated DEREG mice at day 20 pi. C) Frequency of KLRG-1+ and CD25+ Tconv cells in the liver of

PBS or DT-treated DEREG mice at day 11 pi. All data are presented as mean ± SEM. Each symbol represents one individual mouse. Data were pooled from 2–4 independent experiments. Statistical significance was determined by Unpaired t test or Mann Whitney test, according to data distribution. P values for pairwise comparisons are indicated in the graphs. (PDF)

**S8 Fig. Upregulation of CD39 in different cell subsets during *T. cruzi* infection and quantification of ATP and ADO in tissue from Treg cell-depleted infected mice. A-C)** Representative flow cytometry plots (A), frequency (B) and absolute numbers (C) of CD39high Tconv and non-CD4+ cells in the spleen of DEREG mice at day 7 pi (INF) and non-infected controls (NI). **D)** Geometric mean of CD39 fluorescence intensity in CD39+ cells within Treg, Tconv and non-CD4+ T cells in the spleen of NI and INF (7 dpi) DEREG mice. In A-D data were pooled from 3 independent experiments and statistical significance was determined by Unpaired t test or Mann Whitney test, according to data distribution. **E)** Concentration of ATP and Adenosine (ADO) quantified in the supernatants of the spleen, liver and heart of NI, PBS or DT-treated INF (21 dpi) mice. Data in (E) were collected from 1 experiment. Statistical significance was determined by one-way ANOVA followed by Tukey's multiple comparison test. All data are presented as mean ± SEM. Each symbol represents one individual mouse. P values for pairwise comparisons are indicated in the graphs. (PDF)

**S1 Table. List of antibodies used for flow cytometry.** (DOCX)

**S1 Data. Source data for graphs of principal figures in this study.** (XLSX)

**S2 Data. Source data for graphs of supplementary figures in this study.** (XLSX)

## Acknowledgments

We thank MP Abadie, MP Crespo, E Zacca, V Blanco, D Lutti, C Noriega, FA Frontera, SR Oms, RE Villarreal, G Furlán, NM Maldonado, A Romero and L Reyna (Centro de Investigaciones en Bioquímica Clínica e Inmunología) for their excellent technical assistance. We are grateful to C Stempin and F Hellriegel for providing advice for the *in vitro* parasite survival assay. We acknowledge the NIH Tetramer Core Facility for provision of APC and BV421-labeled TSKB20 tetramers and APC-labeled PBS-57 complexed to CD1d tetramers.

## Author Contributions

**Conceptualization:** Eva V. Acosta Rodríguez.

**Formal analysis:** Cintia L. Araujo Furlan.

**Funding acquisition:** Eva V. Acosta Rodríguez.

**Investigation:** Cintia L. Araujo Furlan, Santiago Boccardo, Constanza Rodriguez, Verónica S. Mary, Camila M. S. Gimenez.

**Methodology:** Cintia L. Araujo Furlan, Eva V. Acosta Rodríguez.

**Project administration:** Adriana Gruppi, Carolina L. Montes, Eva V. Acosta Rodríguez.

**Resources:** Simon C. Robson, Adriana Gruppi, Carolina L. Montes, Eva V. Acosta Rodríguez.

**Supervision:** Eva V. Acosta Rodríguez.

**Validation:** Cintia L. Araujo Furlan, Eva V. Acosta Rodríguez.

**Visualization:** Cintia L. Araujo Furlan.

**Writing – original draft:** Cintia L. Araujo Furlan, Eva V. Acosta Rodríguez.

**Writing – review & editing:** Santiago Boccardo, Simon C. Robson, Adriana Gruppi, Carolina L. Montes.

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
