## [Decision Letter · Decision Letter 0]

20 Nov 2023

Dear Dr. Acosta Rodriguez,

Thank you very much for submitting your manuscript "CD39 expression by regulatory T cells drives CD8+ T cell suppression during experimental Trypanosoma cruzi infection" for consideration at PLOS Pathogens. As with all papers reviewed by the journal, your manuscript was reviewed by members of the editorial board and by three independent reviewers. In light of the reviews (below this email), we would like to invite the resubmission of a significantly-revised version that takes into account the reviewers' comments.

This manuscript presents some interesting data and the reviewers feel that the study has been well designed and the work carried out to a high standard. However, there are some concerns raised by the reviewers which require further responses before this manuscript can be considered for publication. In particular reviewer 2 has raised points about over interpretation of the data and whether the diphtheria toxin has any direct effect on T. cruzi independently of its use to regulate Treg levels. Please provide responses to the reviewers concerns (in particular those major concerns raised by reviewer 2) and it would be of advantage to directly test the effect of DT on parasites in vitro to exclude such effects from the interpretation of the results.

We cannot make any decision about publication until we have seen the revised manuscript and your response to the reviewers' comments. Your revised manuscript is also likely to be sent to reviewers for further evaluation.

Sincerely,

Martin Taylor

Guest Editor

PLOS Pathogens

Margaret Phillips

Section Editor

PLOS Pathogens

Kasturi Haldar

Editor-in-Chief

PLOS Pathogens

orcid.org/0000-0001-5065-158X

Michael Malim

Editor-in-Chief

PLOS Pathogens

orcid.org/0000-0002-7699-2064

This manuscript presents some interesting data and the reviewers feel that the study has been well designed and the work carried out to a high standard. However, there are some concerns raised by the reviewers which require further responses before this manuscript can be considered for publication. In particular reviewer 2 has raised points about over interpretation of the data and whether the diphtheria toxin has any direct effect on T. cruzi independently of its use to regulate Treg levels. Please provide responses to the reviewers concerns (in particular those major concerns raised by reviewer 2) and it would be of advantage to directly test the effect of DT on parasites in vitro to exclude such effects from the interpretation of the results.

Reviewer's Responses to Questions

**Part I - Summary**

Reviewer #1: This is a very well presented and conducted study that provides mechanistic evidence on the role of Tregs on parasite control and CD8 T cell responses during acute T. cruzi infection. I do have a few comments that may improve data presentation, which I detail below (Part III -minor Issues).

Reviewer #2: This ms presents some interesting findings relating to a potential role for Tregs in the immune response to acute T. cruzi infection in mice. The topic is of interest and importance because there is very little understanding of why host immunity can usually control T. cruzi infection very well, but does not achieve sterile self-cure. This results in the vast majority of human infections being chronic. It is a reasonable hypothesis, given data from other systems, that Tregs could be constraining anti-parasitic CD8 T cells, which are considered to be the principal effectors against T. cruzi. Here the authors use the DEREG mouse system, which allows specific depletion of Tregs at defined time points using diptheria toxin administration. To my knowledge this is the first time this powerful experimental tool has been used to study T. cruzi infection and the authors also combine this model with adoptive transfer techniques to look more specifically at the role of CD39 in Tregs. Thus, the study could be considered fairly cutting edge for the field. The data and text are presented very clearly and for the most part, the experiments appear to have been well designed and executed.

The main weakness of the paper, in my view, is that the importance of Tregs (and expression of CD39 by them) is over-stated given the fairly limited time-frame of most of the data (most read outs are for a single time point around day 20 post-infection) and the generally small or transient effect sizes observed, particularly with respect to parasite loads.

For example, line 25, line 39, line 109 (Treg cells play a critical role), line 163 (Treg cells play a prominent role), line 276 (the crucial role of Treg cells), line 292 (Treg cells play highly relevant functions). The data do show quite clearly that depletion of Tregs results in a transient expansion of one subset of parasite-specific CD8 T cells in the liver and spleen, but this does not appear to make a very strong or consistent difference to parasite loads, perhaps 2 to 3-fold reduction in the blood and heart, sometimes a reduction in the liver (Fig 2 but not Fig 6) and no difference in the spleen around day 20-22. This might be explained by a mildly inhibitory direct effect of diptheria toxin on the parasites, which needs to be tested for. Furthermore, the data show that Treg depletion had no effect if it was initiated on day 11 instead of day 5. The data are definitely a useful contribution to the field, but without knowing whether Treg depletion makes any long term difference to the course of infection other than at this snapshot in time, or on any Chagas disease relevant pathology, it seems unjustified to claim they are critical/crucial etc.

In the title they claim that “CD39 expression by regulatory T cells drives CD8+ T cell suppression”, but this conclusion hinges on the experiment in Fig6 which involved quite an artificial system of adoptively transferring large numbers of immunosuppressive Tregs and I don’t believe there is evidence that the mechanism involves CD39 directly enough to warrant using the term “drives”. Influences or modulates might be more appropriate. Furthermore, it is not clear how CD39 expression actually translates into higher numbers of CTLs in this system and the specific ability to respond to Tc antigen, at least in vitro, in a polyfunctional (IFNg+ TNF+ CD107a+) way (Fig 3) but not in bi- or mono-functional ways (Fig S4). For example, are adenosine levels actually different in the tissues? Is the Treg adoptive transfer effect dose-dependent and how was the number of cells (1 million) to adoptively transfer decided? What happens with T. cruzi infection in the CD39 knockout mice?

Reviewer #3: In this manuscript, Furlan et al investigated the CD39+ T reg cells mediated immune suppression in early stage of experimental T.cruzi infection by regulating the CD8 T cells responses in CD39 dependent manner. Authors showed that depletion of Treg cells have impacted antigen specific CD8 T cell responses by early priming and activation of conventional CD4+ T cells Overall, this is very well written manuscripts and all the results are supported by nicely executed experiments.

**Part II – Major Issues: Key Experiments Required for Acceptance**

Reviewer #1: No further experiments seem to be required to validate study conclusions

Reviewer #2: They should establish at what level diptheria toxin is directly inhibitory to T. cruzi in vitro and develop a better overview of the impact of Treg depletion on parasite load kinetics e.g. a full parasitemia curve, ideally in both the DEREG system (DT vs PBS) and for infections in the CD39 knockout mice.

Reviewer #3: Following are my comments:

1. In abstract, subsequently influencing CD8+ T cells responses (add responses).

2. Introduction, Page 4; line 60:- Immunological help

3. Introduction, Page 5; line 84-85: “Our findings…..” Please rewrite this sentence as it is not very clear.

4. Result section, Page 7; line 133-134: Please re-write the sentence “Importantly, Treg cell depletion was also observed in frequency …”

5. Authors observed reduced parasitemia in liver and heart at day 20 p.i but not in spleen. Any explanation for this?

6. Did author look at author activation markers like CD69 (early marker) and CD38 on parasite specific CD8 T cells (as shown in figure S3C and S3D Figs)?

7. Did author try to look the activation status of parasite specific poly-functional CD8+ T cells (that may differ compared to overall CD8+ T cells) in Treg depleted infected animals compared to control?

8. Page 10, line 204: Among these clusters, we observed six (?????) that displayed….. Please check and rectify the sentence. (Simply state that we observed 6 unique clusters.)

9. Authors define the Treg population based on just FoxP3 expression. Did they consider using CD25 and CD127 as additional marker?

10. I am not much convinced with gating of CD127 and KLRG-1. Can you show FMO control (same for FoxP-3)?

11. Did authors also checked the CTL activity (Perforin, Granzyme etc) of CD*+ T cells in absence of Treg cells?

12. The quality of figure 4 (a,b & c) is not very good and convincing. Similarly Figure 6 a is not clear.

13. Did authors try to block CD39 in vivo (apart from adoptive transfer experiment)?

**Part III – Minor Issues: Editorial and Data Presentation Modifications**

Reviewer #1: 1- Presumably, authors have collected phenotypic data on Tregs from naive DEREG mice (i.e. day 0 post-infection). It would be valuable to add these data to the kinetics graphs from Fig. 1 (C and D).

2- Similarly, it would be valuable to add the kinetics of blood parasitaemias for different groups (PBS and DT) to fig. 2, and not just day 20 pi as currently displayed (Fig. 2A). This would allow a visual comparison of the kinetics of blood parasitaemia vs kinetics of TSKB20+ antigen-specific CD8 T cell responses as a consequence of Treg depletion (i.e. Same time points for Fig. 2 A and D).

3- Fig 3, it would be valuable to show the FMO control for the CD127 staining

4- Fig 3, the impact of Treg depletion on antigen-specific CD8 T cell responses are very limited in terms of absolute counts (i.e. only counts of TSKB20+ SLECs in the liver are affected). Authors should further discuss this apparently limited impact.

5- Fig 6A, contour plots with outliers instead of density plots will probably be a better representation for this data. Actually, replacing density plots with contour plots (or zebra plots) throughout the manuscript will probably improve flow data presentation. In addition, displaying FMO controls for CTLA-4, CD25 and CD39 in this figure would be of value.

6- Fig 6B, what about absolute counts? Could authors please display absolute counts side-by-side % in Fig 6B?

7- Fig 6E, have authors checked for viability of cells before adoptive transfer? Please clarify and display data if possible.

8- Fig 6E, authors mention they used in vitro splenocytes from CD39 KO mice to generate KO iTregs. However, the CD39 KO mouse strain used in the study is not mentioned in M&M. Please, add this information.

9- Authors suggest CD39 as a potential target for immunomodulatory therapeutic strategies during T. cruzi infection. However, authors performed this study only during acute infection, and effects were only detected following very early interventions (i.e. 5-6 days pi) after primary infection. Strong word of caution here; it is extremely unlikely for clinicians to intervene very early after primary T. cruzi infection in humans. In any case, even if very early intervention was possible (e.g. in the case of a laboratory accident), the preferred avenue for treatment will certainly be benznidazole, for which there is strong clinical evidence of its benefits during acute infection. Therefore, I would recommend not to suggest CD39 as a potential therapeutic strategy at least until after authors are able to collect meaningful data during chronic experimental infections.

Reviewer #2: 1. For some data sets the legends state that data are pooled from independent experiments while for others the authors show data from one experiment and state that similar results were obtained in independent experiments. How did they decide when to pool data and when to not pool? Does “similar results” mean that the same statistically significant differences were reproduced in all cases?

2. Lines 59-61: The three papers cited here are all on immune responses in mice and come from the same lab – have these phenomena been shown to be reproducible in human studies?

3. Line 81-88: Should clarify that the study discussed relates to mice. In general, the introduction would benefit from a stronger distinction between mouse and human data, even if it means highlighting where there are gaps regarding human infections.

4. Figure 1C,D,E: It is difficult to interpret these figures because the data are presented as the frequency of Tregs within the total CD45+ population, which is going to be massively affected by increasing frequencies of other cell types e.g. CD8 T cells, as a natural consequence of the infection. It would be clearer to present the frequencies as % of total CD4+ cells rather than/as well as CD45+ cells. The authors do present total numbers of Tregs, but only for spleen and liver at one time point (day 20) and here the depletion is significant, but with quite a small effect size.

5. Related to this, was there a difference in spleen and liver weight between the PBS and DT-treated infected groups? Was this taken into account?

6. Figure 2B and 6G: y axis label

---

## [Decision Letter · Decision Letter 1]

12 Apr 2024

Dear Dr. Acosta Rodriguez,

We are pleased to inform you that your manuscript 'CD39 expression by regulatory T cells participates in CD8+ T cell suppression during experimental Trypanosoma cruzi infection' has been provisionally accepted for publication in PLOS Pathogens.

Best regards,

Martin Taylor

Guest Editor

PLOS Pathogens

Margaret Phillips

Section Editor

PLOS Pathogens

Michael Malim

Editor-in-Chief

PLOS Pathogens

orcid.org/0000-0002-7699-2064

Reviewer Comments (if any, and for reference):

Reviewer's Responses to Questions

**Part I - Summary**

Reviewer #1: The authors did an excellent job addressing reviewers’ concerns. They even produced additional experimental data. This is now a very solid piece of work. I believe this is ready to be accepted for publication.

Reviewer #2: The authors have done substantial additional experiments and analyses that improve the strength of their conclusions. They also modified the text in important ways to clarify interpretation, add more context and cover more of the nuances in their data sets. They rebutted a few of my more minor points and I accept their justifications as a difference of opinion. Overall, the new version is much improved and I have no further concerns.

Reviewer #3: Authors have clarified all the concerns raised by me.

**Part II – Major Issues: Key Experiments Required for Acceptance**

Reviewer #1: (No Response)

Reviewer #2: (No Response)

Reviewer #3: N/A

**Part III – Minor Issues: Editorial and Data Presentation Modifications**

Reviewer #1: (No Response)

Reviewer #2: (No Response)

Reviewer #3: N/A

PLOS authors have the option to publish the peer review history of their article (what does this mean?). If published, this will include your full peer review and any attached files.

Reviewer #1: **Yes: **Damián Pérez-Mazliah

Reviewer #2: No

Reviewer #3: No

---

## [Editor Report · Acceptance letter]

24 Apr 2024

Dear Dr. Acosta Rodriguez,

We are delighted to inform you that your manuscript, "CD39 expression by regulatory T cells participates in CD8+ T cell suppression during experimental *Trypanosoma cruzi* infection," has been formally accepted for publication in PLOS Pathogens.

Best regards,

Michael Malim

Editor-in-Chief

PLOS Pathogens

orcid.org/0000-0002-7699-2064